# Diminished responses to bodily threat and blunted interoception in suicide attempters

Danielle C DeVille[1,2], Rayus Kuplicki[1], Jennifer L Stewart[1,3], Tulsa 1000 Investigators[1], Martin P Paulus[1,3], Sahib S Khalsa[1,3]*

[1]Laureate Institute for Brain Research, Tulsa, United States; [2]Department of Psychology, The University of Tulsa, Tulsa, United States; [3]Oxley College of Health Sciences, The University of Tulsa, Tulsa, United States

**Abstract** Psychological theories of suicide suggest that certain traits may reduce aversion to physical threat and increase the probability of transitioning from suicidal ideation to action. Here, we investigated whether blunted sensitivity to bodily signals is associated with suicidal action by comparing individuals with a history of attempted suicide to a matched psychiatric reference sample without suicide attempts. We examined interoceptive processing across a panel of tasks: breath-hold challenge, cold-pressor challenge, and heartbeat perception during and outside of functional magnetic resonance imaging. Suicide attempters tolerated the breath-hold and cold-pressor challenges for significantly longer and displayed lower heartbeat perception accuracy than non-attempters. These differences were mirrored by reduced activation of the mid/posterior insula during attention to heartbeat sensations. Our findings suggest that suicide attempters exhibit an 'interoceptive numbing' characterized by increased tolerance for aversive sensations and decreased awareness of non-aversive sensations. We conclude that blunted interoception may be implicated in suicidal behavior.

**\*For correspondence:**
skhalsa@laureateinstitute.org

**Group author details:**
Tulsa 1000 Investigators See page 14

**Competing interests:** The authors declare that no competing interests exist.

## Introduction

Suicide ranks among the leading causes of death worldwide (*World Health Organization, 2014*). In the US alone, suicide increased by nearly 30 percent between 2000 and 2016 (*World Health Organization, 2014*; *Hedegaard et al., 2018*). For every death by suicide, it is estimated that there are 25 additional suicide attempts (*Hedegaard et al., 2018*), each associated with significant social, emotional, and financial consequences. Experts have strived to understand and prevent death by suicide for decades, and yet, our current scientific grasp of the factors that contribute to suicidal behavior is lacking. Moreover, epidemiological data suggest that we are no better at preventing death by suicide than we were 100 years ago (*Hedegaard et al., 2018*; *United States Department of Commerce Bureau of the Census, 1920*; *Kessler et al., 2005*; *Nock et al., 2008*), with suicide rates rising despite the application of prevention and intervention efforts (*Linehan, 2008*; *Nock, 2016*; *Paris, 2006*).

Theoretical models of suicide have invoked the concept of 'suicidal capacity' to differentiate the small subset of individuals who attempt suicide from the much larger group of individuals who experience suicidal ideation but never resort to suicidal action (*Ribeiro and Joiner, 2009*; *Smith and Cukrowicz, 2010*). A basic tenet of this concept is the notion that most human beings are 'hard-wired' for survival and thus driven to avoid physical pain and threats to bodily homeostasis. Psychological theories of suicide suggest that in an individual with heightened suicidal capacity, certain dispositional (*Klonsky and May, 2015*) and acquired (*Van Orden et al., 2010*; *Van Orden et al., 2008*) traits result in a lower aversion to physical threat and a higher likelihood of transitioning from

**eLife digest** The human brain closely monitors body signals essential for our survival, including our heartbeat, our breathing and even the temperature of our skin. This mostly unconscious process is called interoception. It helps people perceive potential or actual threats and helps them to respond appropriately. For example, a person charged by a wild animal will act instinctively to run, fight or freeze. Unlike most creatures, humans show an ability to counteract these survival instincts, and are capable of intentionally engaging in behaviors that result in physical harm. Recent increases in the rate of suicide have made it more urgent to try to understand what leads to this behavior in humans.

Now, DeVille et al. show that people with psychiatric disorders who have survived a suicide attempt have blunted interoception. In four experiments, people with a history of suicide attempts were compared to another group of individuals without a history of suicide attempts. The groups were carefully matched such that there were no significant differences in the demographic and clinical characteristics of the two groups, including in terms of their age, sex, body mass index and psychiatric symptoms.

Both groups completed uncomfortable tasks like holding their breath or keeping their hand in icy cold water. The participants also completed two tasks that required them to focus on their own heartbeat, one of which was paired with functional magnetic resonance imaging. Those with a history of suicide attempts held their breath and kept their hand in cold water for longer, and also were less in tune with their heart rate. This "interoceptive numbing" was associated with less activity in part of the brain called the insular cortex. These differences could not be explained by the individuals having a psychiatric disorder or a history of considering suicide, or by them taking psychiatric medications. Instead, the interoceptive numbing was most often seen in individuals who made an attempt on their own life.

The experiments identify physical characteristics that may differentiate people who attempt suicide from those who do not. This lays the groundwork for future research aimed at identifying biological indicators of suicide risk. More studies are needed to verify the results. If the results are verified, the next step would be prospective studies to determine whether measuring interoception can help clinicians predict who is at risk of a suicide attempt. If it does, it might give clinicians a new tool to try to prevent suicide by ensuring those at greatest risk receive appropriate care.

suicidal ideation to action. Consistent with this line of thinking, non-suicidal self-injury (*Klonsky and May, 2014*; *Franklin et al., 2011*) and high levels of fearlessness of the pain involved in dying (*May et al., 2012*) are behavioral and clinical factors that have been reported to predict suicide attempts. Thus it seems possible that suicidal behavior might be influenced by one's ability to access and respond adaptively to homeostatic information regarding the internal state of the body, but few studies have directly investigated this topic.

Interoception describes the nervous system's process of sensing, interpreting, and integrating signals originating from inside the body (*Craig, 2002*; *Khalsa et al., 2018*). Emerging evidence suggests that dysfunctions of interoception may contribute to certain mental illnesses (*Khalsa et al., 2018*; *Khalsa and Lapidus, 2016*), including mood and anxiety disorders (*Paulus and Stein, 2010*; *Avery et al., 2014*; *Barrett et al., 2016*; *Wiebking et al., 2015*; *Harshaw, 2015*; *Domschke et al., 2010*), substance use disorders (*Paulus and Stewart, 2014*; *Verdejo-Garcia et al., 2012*), eating disorders (*Kerr et al., 2016*; *Berner et al., 2018*; *Khalsa et al., 2015*), and nonsuicidal self-injury (*Muehlenkamp, 2012*), all of which are associated with an elevated risk of suicide (*Nock et al., 2010*; *Harris and Barraclough, 1997*; *Smith et al., 2018a*). Interoception is thought to be substantially supported by the insular cortex, with the primary representation of visceral sensations occurring in the mid-to-posterior insula, and the integration of interoceptive information with cognition, emotion, and other higher order processes occurring in more anterior regions (*Barrett and Simmons, 2015*; *Critchley and Harrison, 2013*; *Hassanpour et al., 2018*).

To test the hypothesis that abnormalities of interoception are associated with suicidal capacity in individuals with psychiatric disorders, we evaluated interoceptive processing in participants with a history of suicide attempts as compared to a matched psychiatric reference sample of participants

with no history of suicide attempts. We measured aversive interoceptive processing across the respiratory and nociceptive domains, via an inspiratory breath-hold challenge and a cold-pressor challenge. We assessed cardiac interoception during a heartbeat perception task as well as during a functional magnetic resonance imaging (fMRI) task involving focused attention to heartbeat sensations. We predicted that relative to non-attempters, suicide attempters would 1) tolerate aversive interoceptive sensations to a greater extent, 2) demonstrate lower interoceptive accuracy, and 3) exhibit differences in brain activity in the insular cortex when attending to interoceptive sensations.

## Results

### Demographic and clinical characteristics

We found that both participant groups were well-matched in terms of demographic and clinical characteristics, showing no significant differences in age, BMI, sex, diagnosis, or levels of self-reported depression, anxiety, substance use, or eating disorder symptoms (*Table 1*). We noticed that the groups showed a significant difference in their usage of psychotropic medication, with a greater proportion of the suicide attempters reporting taking such medications. We provide further details regarding our participants, including psychiatric diagnoses, use of psychotropic medications, missing data values, and scores on self-report measures in Appendix 1.

### Breath-hold challenge

We found that suicide attempters held their breath for significantly longer than non-attempters, approximately 10 s longer on average across both trials (F (1, 121.84) = 4.48, p = 0.036, $R^2$ = 0.042) (see *Figure 1*). We also observed a repetition effect, such that all participants held their breath longer during the second trial (F(1,97.01) = 20.18, p < 0.001, $R^2$ = 0.173), replicating previous results with this task (*Willem Van der Does, 1997*). We did not find a significant interaction between group and trial. We report a summary of the Linear Mixed Effects (LME) output for the model examining breath-hold duration, including fixed effects estimates and standardized regression coefficients in *Supplementary file 1*.

Concordant with the increased breath-hold duration in suicide attempters, we found that suicide attempters had higher concentrations of exhaled carbon dioxide ($CO_2$) than non-attempters after the breath-hold trials (F(1,120.33) = 5.52, p < 0.001, $R^2$ = 0.043). However, we did not find an effect

**Table 1.** Participant demographics and clinical characteristics.

| | Suicide attempters (*n* = 34) | Non-attempters (*n* = 68) | p |
|---|---|---|---|
| Demographics | | | |
| Age, years | 31 (11) | 33 (10) | 0.40 |
| BMI | 27.1 (6.3) | 28.6 (5.0) | 0.20 |
| % Female | 56 (n = 19) | 70 (n = 47) | 0.27 |
| Clinical Features | | | |
| PHQ-9 | 12.1 (5.4) | 11.2 (5.7) | 0.42 |
| PHQ-9 SI question | 0.5 (0.8) | 0.3 (0.5) | 0.12 |
| OASIS | 9.2 (4.6) | 8.5 (4.2) | 0.47 |
| DAST | 3.7 (4.2) | 3.5 (4.0) | 0.89 |
| SCOFF | 1.7 (1.6) | 1.3 (1.4) | 0.16 |
| % Medicated | 85 (n = 29) | 59 (n = 40) | 0.01 |

*Note:* All values reported are in the format of *Mean (SD)* unless otherwise indicated.

BMI = Body Mass Index; PHQ-9 = Patient Health Questionnaire; SI = Suicidal Ideation; OASIS = Overall Anxiety Severity and Impairment Scale; DAST = Drug Abuse Screening Test; SCOFF = Eating Disorders Screening Tool. All scores on clinical measures reflect total scores unless otherwise specified. For all clinical measures, higher numbers indicate greater endorsement of the construct assessed.

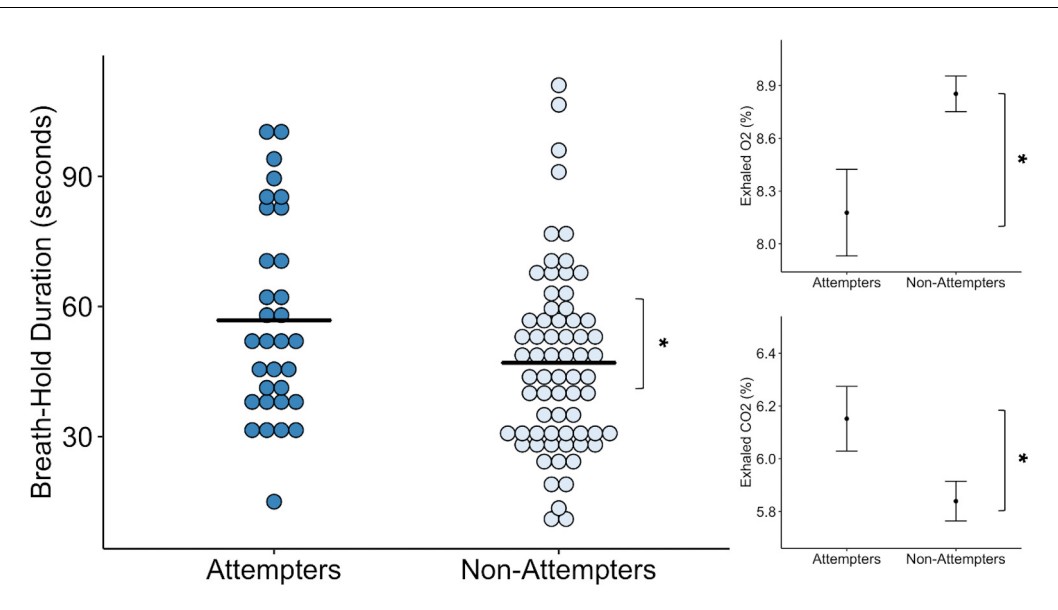

**Figure 1.** Suicide attempters held their breath significantly longer than non-attempters during the inspiratory breath-hold challenge (approximately 10 s on average). They also exhibited greater increases in carbon dioxide ($CO_2$) and decreases in oxygen ($O_2$). The mean breath-hold duration across the two trials is displayed below. Error bars indicate + / - 1 standard error, *p < 0.05.

The online version of this article includes the following source data and figure supplement(s) for figure 1:

**Source data 1.** Processed breath-hold task data.
**Figure supplement 1 .** Suicide attempters and non-attempters exhibited no differences in symptom ratings following the breath hold task.

---

of trial or interaction between group and trial. We also found that suicide attempters had lower concentrations of exhaled oxygen ($O_2$) following the breath-hold trials relative to non-attempters ($F_{(1,132.27)} = 5.00$, $p = 0.027$, $R^2 = 0.036$). We observed a significant main effect of trial ($F_{(1,91.09)} = 6.16$, $p = 0.015$, $R^2 = 0.020$), such that reductions in $O_2$ were greater after the second breath-hold across both groups. We report summaries of the LME outputs for the $O_2$ and $CO_2$ models, including fixed effects estimates and standardized regression coefficients in ***Supplementary file 1***.

Despite the prolonged breath-hold duration and elevations in $CO_2$, we found that suicide attempters did not report any differences in perceived breathlessness (p = 0.70), feelings of suffocation (p = 0.95), fear of suffocation (p = 0.97), urge to breathe (p = 0.76), breathing sensation intensity (p = 0.53), unpleasantness (p = 0.63), task difficulty (p = 0.48), or effort expended during the breath-hold (p = 0.27) relative to non-attempters (***Figure 1—figure supplement 1*** ).

## Cold-pressor challenge

We found that the cold-pressor challenge elicited increased pain ratings over time in both groups ($F_{(3,276.16)} = 86.78$, $p < 0.001$, $R^2 = 0.589$). However, this effect was qualified by a significant interaction between timepoint and group ($F_{(3,277.39)} = 2.89$, $p = 0.036$, $R^2 = 0.030$). On closer examination of the LME fixed effects, we observed that suicide attempters kept their hands submerged in the cold water for significantly longer than non-attempters after reaching their peak pain level ($t_{(278.56)} = 2.78$, $p = 0.006$, $\beta = 0.13$), without any significant differences in the amount of time taken to reach mild, moderate, and peak pain levels (***Figure 2***). Overall, suicide attempters kept their hands submerged in the icy water for approximately 18 s longer than the non-attempters. We report fixed effects and model summary values in ***Supplementary file 2***. Additionally, although suicide attempters provided slightly lower average ratings of unpleasantness, pain, difficulty, and stress than non-attempters, these differences were not statistically significant (unpleasantness: U = 1107,

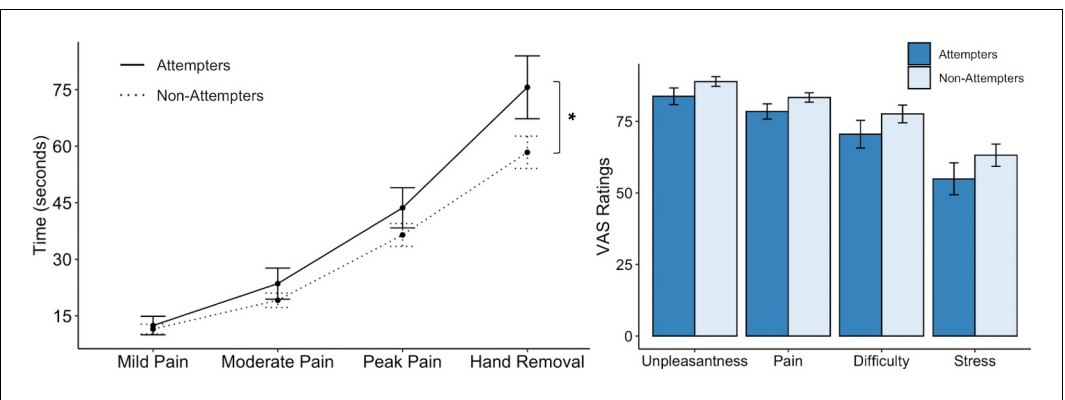

**Figure 2.** Suicide attempters exhibited significantly greater pain tolerance than non-attempters during the cold-pressor challenge. However, they did not significantly differ in their retrospective ratings of overall pain, unpleasantness, difficulty, or stress experienced during the task. Error bars indicate + / - 1 standard error; *p < 0.05.

The online version of this article includes the following source data for figure 2:

**Source data 1.** Processed cold-pressor challenge data.

p = 0.117, FDR-p = 0.144, r = 0.16; pain: U = 1067, p = 0.122, FDR-p = 0.144, r = 0.15; difficulty: U = 1123, p = 0.090, FDR-p = 0.144, r = 0.17; stress: U = 1095, p = 0.144, FDR-p = 0.144, r = 0.15).

## Heartbeat perception task

Our initial LME model examining heartbeat perception accuracy as a function of group, condition (i.e. guess, no-guess, and perturbation), and their interaction, showed a significant effect of condition (F(2,195.08) = 12.72, p < 0.001, $R^2$ = 0.200). However, there was no significant effect of group and no group by condition interaction. By examining the fixed effects we noticed that, relative to guessing trials, accuracies on the no-guess (t(195.08) = −4.61, p < 0.001, β = −0.33) and breath-hold perturbation trials (t(195.07) = −4.013, p < 0.001, β = −0.29) were significantly lower.

We made a post-hoc decision to apply a second model that omitted the guessing score from the analysis, based on a recent study indicating that heartbeat perception accuracy scores are potentially confounded by guessing (*Desmedt et al., 2018*). For the second heartbeat perception model, we examined accuracy as a function of group and condition across the no-guess and breath-hold perturbation trials only (i.e. after omitting the 'guess' trial), and found a significant difference between groups (F(1,97.04) = 8.64, p = 0.004, $R^2$ = 0.048) (*Figure 3*) and a significant interaction between group and trial (F(1,144.47) = 4.37, p = 0.04, $R^2$ = 0.043). In particular, we found that suicide attempters exhibited lower heartbeat perception accuracy during the no-guess condition relative to non-attempters, t(144.46) = −2.94, p = 0.003, β = −0.29), and that the difference in accuracy between attempters and non-attempters was attenuated during the perturbation trial (t(97.73) = 2.09, p = 0.04, β = 0.14). We did not observe group differences in ratings of task confidence or difficulty across the no-guess and perturbation trials (Confidence: U = 1153.5, p = 0.453, FDR-p = 0.67, r = 0.08; Difficulty: U = 1014.5, p = 0.766, FDR-p = 0.767, r = 0.03). Suicide attempters displayed a tendency to rate their heartbeat sensations as less intense (U = 1345, p = 0.028), although this was non-significant after applying a Benjamini-Hochberg correction across contrasts (FDR-p = 0.084, r = 0.22) (*Figure 3*). We report fixed effects and model summary values in *Supplementary file 3*.

## Interoceptive attention task during fMRI

We also found that, relative to non-attempters, suicide attempters exhibited reduced BOLD activation in the right dorsal mid-insula and right posterior insula during interoceptive attention to the heartbeat versus the exteroceptive attention condition (p < 0.005, corrected at α <0.05; *Figure 4*). There was also a cluster of reduced BOLD activation within the left dorsal mid-insula, but this did not survive correction. The whole-brain analysis revealed four additional clusters with significantly reduced BOLD activation during attention to heart sensations among suicide attempters: one cluster within the right precuneus, one within the right superior temporal gyrus, one within the right

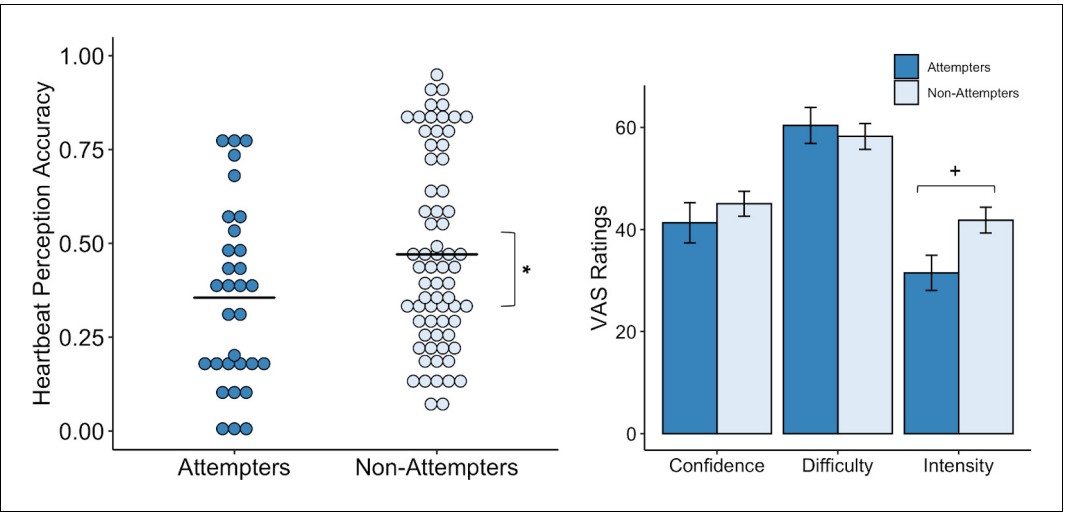

**Figure 3.** Suicide attempters exhibited significantly lower heartbeat perception accuracy than non-attempters during the no-guess and breath-hold perturbation conditions. There were no perceived differences in reported in task difficulty or confidence in performance across the no-guess and perturbation conditions. Suicide attempters also provided lower ratings of heartbeat intensity across these conditions, although this was no longer significant after correction for multiple comparisons. Error bars indicate + / - 1 standard error; *p < 0.05, +p < 0.10. The online version of this article includes the following source data for figure 3:

**Source data 1.** Processed heartbeat perception task data.

posterior cingulate cortex, and one within the right dorsomedial prefrontal cortex (p < 0.0005, ACF corrected at α < 0.05; *Table 2*).

## Correlations across interoceptive tasks and measures

We report exploratory correlations across all behavioral and neuroimaging variables in *Appendix 1—figure 1*.

## Role of psychotropic medication status

We conducted additional analyses to examine potential confounding effects of medication status on our primary interoception variables of interest, due to the statistically significant difference observed in the proportion of individuals taking psychotropic medications in each group. Our results remained largely unchanged after accounting for medication status, as detailed in Appendix 2.

## Role of suicidal ideation

We did not initially account for the role of suicidal ideation in the current study, focusing instead on interoceptive processing differences between individuals with a history of suicide attempts within the last 5 years and individuals with no suicide attempt history. To rectify this issue, we conducted additional analyses examining whether suicidal ideation history might explain the observed abnormalities of interoception across subjective, behavioral, and neural levels. Our observation of diminished interoception in suicide attempters was largely unchanged after accounting for lifetime suicidal ideation intensity, as detailed in Appendix 2 and displayed in *Appendix 2—figure 1*.

## Discussion

We investigated whether attenuated interoceptive processing is associated with self-reported suicide attempts in individuals with a range of psychiatric disorders including depression, anxiety, post-traumatic stress disorder, eating disorders, and/or substance use disorders. We found that suicide attempters show reduced responses to homeostatic threats to the body, including increased tolerance for sensations of air hunger and increased tolerance of cold pain relative to non-attempters. Additionally, we found that suicide attempters exhibit decreased heartbeat perception accuracy and

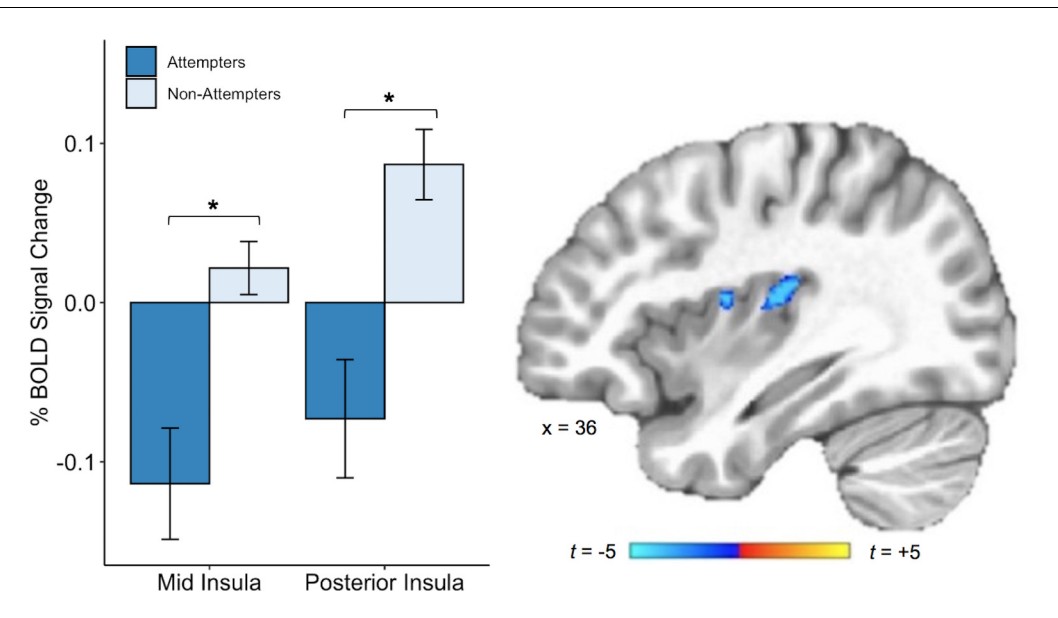

**Figure 4.** Suicide attempters exhibited lower blood oxygen level-dependent (BOLD) signal in the right posterior and mid insula than non-attempters during attention to heartbeat sensations (relative to the exteroceptive condition; p < 0.005, ACF corrected at α < 0.05). Error bars indicate + / - 1 standard error.

The online version of this article includes the following source data for figure 4:

**Source data 1.** Subject-level beta values from mid and posterior insula cluster activation during interoceptive (relative to exteroceptive) conditions.

decreased mid and posterior insula activity when attending to sensations from the heart, an interoceptive organ that is vital for maintaining survival. Taken together, these findings provide initial support for the hypothesis that an increased capacity to engage in self-destructive and life-threatening behaviors is associated, behaviorally and neurobiologically, with a blunted sensitivity to internal bodily signals.

Prior to the current investigation, only a few empirical studies have examined the relationship between suicidality and interoception. These studies also suggested a role for interoceptive deficits in suicidal behavior, although the measurement of interoception was restricted solely to self-report questionnaires (*Forrest et al., 2015*; *Dodd et al., 2018*; *Smith et al., 2018b*; *Rogers et al., 2018*). Here, using a battery of behavioral measures of interoceptive processing, we demonstrate that interoceptive dysfunction in suicide attempters extends beyond symptom measures and includes

**Table 2.** Brain regions exhibiting significantly decreased blood oxygen level-dependent (BOLD) signal during attention to heartbeat sensations relative to exteroceptive sensations in suicide attempters relative to non-attempters

| Location | MNI Coordinates | | | Peak t | Volume (mm³) |
|---|---|---|---|---|---|
| | x | y | z | | |
| Right Dorsal Posterior Insula | 31 | −19 | 15 | −3.7 | 469 |
| Right Dorsal Mid-Insula | 39 | -1 | 13 | −3.4 | 352 |
| Right Precuneus and Posterior Cingulate | 5 | −61 | 27 | −4.8 | 1568 |
| Right Superior Temporal Gyrus | 65 | −29 | 5 | −5.2 | 784 |
| Right Middle Cingulate Cortex | 11 | -7 | 45 | −5.0 | 712 |
| Right Superior Medial Gyrus | 1 | 57 | 17 | −5.2 | 544 |

Note: A voxel-wise threshold of p < 0.005 was set for the insula, and a voxel-wise threshold of p < 0.0005 for the rest of the brain; all significant activations passed a cluster-size correction for multiple comparisons of α < . 05.

abnormalities across behavioral, physiological, and neural indices. Furthermore, we observed these differences in a well-characterized sample of participants with similar levels of psychiatric symptoms, increasing the likelihood that the history of suicidal behavior was the driving force behind the differences observed in interoception rather than a varying expression of psychopathology between groups.

Our use of the cold-pressor and inspiratory breath-hold challenges enabled us to examine interoceptive responses to homeostatic threat, revealing that suicide attempters were able to sustain both tasks for a longer period of time than non-attempters. There are several potential explanations for these findings. Suicide attempters may be less sensitive to the physiological cascade that typically follows a homeostatic threat to bodily integrity, allowing them to persist at the task for longer before noticing and responding to internal physiological cues to withdraw. Indeed, the suicide attempters demonstrated an ability to continue the breath-hold task for longer time periods than non-attempters, even in the face of greater reductions in $O_2$ and increases in $CO_2$ (i.e. physiological indicators of potential bodily harm). Suicide attempters may also appraise signals of homeostatic threat (e.g. pain, breathlessness) as less salient, resulting in slower withdrawal from aversive homeostatic perturbations. For example, while the suicide attempters sustained breath-hold and cold-pressor perturbations longer than to non-attempters, they did not demonstrate the expected corresponding increases in unpleasantness, stress, difficulty, and other ratings of task aversiveness. Rather, in the cold-pressor challenge, the suicide attempters' ratings of stress and difficulty were marginally *lower* than that of non-attempters, even though they continued the cold-pressor challenge for an average of 18 s longer. Based on the current findings, we speculate that a disconnection between the physiological and affective experiences of pain may enable such individuals to engage in self-injurious behaviors and to fail to withdraw from painful stimuli, possibly to the point of inflicting physical harm.

Suicide attempters, relative to non-attempters, also displayed reduced interoceptive accuracy for heartbeat sensations and decreased hemodynamic responses in the right dorsal mid and posterior insula during interoceptive attention to naturally-occurring heart sensations. In terms of the importance of these regions, the mid/posterior insula is often presumed to be the primary recipient of thermal, nociceptive, tactile, and cardiovascular inputs from the ventromedial thalamic nuclei (*Craig, 2002*; *Critchley and Harrison, 2013*), supporting the subsequent representation of conscious awareness of internal bodily states. Prior research suggests that the dorsal mid-insula is sensitive to homeostatic signals (*Simmons et al., 2013b*) and critical to the brain's representation of visceral stimulation (*Hassanpour et al., 2018*), visceral attention (*Farb et al., 2013*; *Simmons et al., 2013a*), and interoceptive memory (*DeVille et al., 2018*). From a clinical perspective, dysfunction in the dorsal mid-insula has been linked to somatic symptoms (*Avery et al., 2014*) and failure to meet energy needs (*Simmons et al., 2016*) among individuals with depression, suggesting that functional abnormalities within this region may be associated with the inability to effectively attend to and use homeostatically relevant information from the body. Additionally, the posterior insula has been conclusively linked to the detection of aversive sensory stimuli and associated shifts in behavioral response strategies in mice (*Gehrlach et al., 2019*), and it is a brain region that is closely tied to the amygdala and threat processing (*Berret et al., 2019*; *Livneh et al., 2017*). Meta-analytic functional neuroimaging findings in humans have specifically implicated the mid/posterior insula in cardiac interoception, with a particular role in attending to and accurately perceiving naturally occurring cardiac sensations (*Schulz, 2016*). Collectively, these studies illustrate that the observed blunting of insular cortex activation in suicide attempters is precisely located within subregions that are closely relevant for aversive threat processing, conscious awareness of the heartbeat signal, and homeostatic regulation.

Prior theoretical work has suggested that a low signal-to-noise ratio of visceral input to the insula may influence interoception among individuals with certain psychiatric illnesses, who are thought to be less capable of discriminating random signal fluctuations from biologically relevant signals (*Paulus and Stein, 2010*; *Barrett et al., 2016*; *Barrett and Simmons, 2015*). As a result, it is thought that these individuals tend to erroneously evaluate benign signal changes as significant, and plan and act on these signals accordingly (*Paulus et al., 2019*). In this view, the suicide attempters' reduced mid and posterior insula activation during heartbeat interoception could reflect a 'noisier' or weaker processing of interoceptive afferents, potentially interfering with the ability to accurately detect signals from their bodies. To extend this line of thought, it is also possible that difficulty

distinguishing signal from noise manifests as a reduced ability to adaptively detect homeostatically-relevant signals (e.g. those related to pain and/or suffocation, as in the cold-pressor and breath-hold tasks), which could help to explain the group differences in homeostatic threat response observed in the current study. However, these notions remain tentative until further research can replicate and extend the findings across other measures of response to aversive and/or painful interoceptive stimuli.

We found that suicide attempters exhibited reduced activation within cortical midline structures during cardiac interoceptive attention (i.e. dorsomedial prefrontal cortex, posterior cingulate, and precuneus) that have been previously implicated as core brain regions underlying human self-referential processing (*Northoff and Bermpohl, 2004*; *Northoff et al., 2006*; *Lemogne et al., 2011*). Our finding is consistent with prior research linking abnormal self-referential processing to suicidal ideation and behavior (*Marchand et al., 2012*) and suggests that further exploration for a role of interoception in the relationship between interoception and self-referential processing in suicide may be worthwhile. Suicide attempters also exhibited reduced activation within the superior temporal gyrus during interoceptive attention. Prior studies have linked suicide attempts with structural and functional abnormalities within the superior temporal gyrus (*Soloff et al., 2012*; *Pan et al., 2015*; *Aguilar et al., 2008*; *van Heeringen et al., 2014*), arguing for a role in socio-emotional threat evaluation, but further investigation would be necessary to pinpoint the role of interoception in this relationship.

Our findings provide some key empirical support for modern theoretical models of suicide that are built upon desire-capability frameworks (*Klonsky et al., 2017*), such as the motivational-volitional model (*O'Connor and Kirtley, 2018*), the interpersonal-psychological theory of suicide (*Ribeiro and Joiner, 2009*; *Van Orden et al., 2010*), and the three-step theory (*Klonsky and May, 2015*). One assertion shared by each of these models is that, for suicide attempts and/or death by suicide to occur, the suicidal individual must express the capacity to approach (rather than avoid) bodily harm. According to these theories, individuals who engage in suicidal behavior demonstrate an ability to ignore—or override—the natural aversion to bodily harm that protects humans against physical injury. Our observations are consistent with these theories and provide early experimental evidence for the role of blunted interoception and heightened tolerance for homeostatic threat in suicidal attempts.

## Limitations and future directions

Although our study represents the most comprehensive investigation related to interoception and suicide to date, we must acknowledge certain limitations. We evaluated evidence for interoceptive processing focusing on individuals with a history of suicide attempts within the last 5 years, based on the report that self-reported interoceptive deficits may be greater among individuals with more recent suicide attempts (*Forrest et al., 2015*). An alternative approach for future research might be to compare performance on neural and behavioral constructs related to interoception in individuals with more recent suicide attempts. Another limitation is that, while our findings suggest that individuals with suicide attempts exhibit abnormal interoception, we did not fully examine whether a history of suicidal ideation—versus a suicide attempt—has an independent impact on interoception. To begin to address this point, we conducted additional analyses which suggested that the observation of diminished interoception in suicide attempters was largely unchanged after accounting for lifetime suicidal ideation. But prospective studies are needed to conclusively discern whether the relationship between interoception and suicide attempt history can be attributed to group differences in suicidal ideation. Additionally, after matching our suicide attempter and non-attempter samples on measures of psychopathology, we found that the proportion of participants taking psychotropic medications at the time of data collection was significantly greater in the suicide attempters. Accounting for these differences in subsequent analyses did not substantially affect our results. One possibility is that the greater psychotropic medication usage in this group might reflect an effort by clinicians to reduce further suicide attempts.

From our cross-sectional study, it is difficult to judge whether the observed differences in interoception represent predispositions (i.e. innate characteristics), whether they reflect an emerging response at some point during the development of suicidal ideation, or occur as a response to suicidal behavior. Addressing these questions via longitudinal task-based assessments of interoception and/or pain processing would provide valuable insight into the impact of blunted interoception on

the emergence of suicidal ideation and the conversion to suicidal behavior (*Millner et al., 2017*). Although not the primary intent of our investigation, we observed several interrelationships within and across levels of analysis raising the possibility of a latent interoceptive awareness trait factor. However, these relationships were inconsistent and were not pre-specified in our hypotheses. Identification of such a latent factor would likely require additional investigation using larger samples and inclusion of individuals not meeting criteria for psychiatric disorders.

We also used an imprecise, albeit commonly employed, measure of pain perception in the cold-pressor challenge. It would be advantageous to clarify whether suicidal action is differentially linked to impaired processing of other pain signals. Examples include visceral pain, which tends to be poorly localized, often referred to somatic structures and produces strong autonomic and affective responses, as well as other somatic pain signals (e.g. thermal or mechanical pain), which tend to be discretely localized to somatic structures and produce more variable autonomic and affective responses (*Sikandar and Dickenson, 2012*). Beyond stimulating visceral and somatic pain processing via different neuroanatomical pathways, it would be helpful to evaluate the degree to which altered pain responding is directly driven by differences in nociception per se as opposed to indirectly modulated by differences in interoceptive processing (*Pollatos et al., 2012*). Additionally, it is increasingly understood that cardiac interoception is rather difficult to assess (*Khalsa and Lapidus, 2016*). Heartbeat perception tasks such as the one employed in the current study are widely used, but have been the subject of criticism (*Desmedt et al., 2018*), and there is evidence to suggest that performance on this type of task can be influenced by one's a priori knowledge about their heartbeat (*Murphy et al., 2018*). We addressed some of these potential confounds in our heartbeat perception task by including a no-guess trial condition, and a trial in which an inspiratory perturbation was used to putatively increase the intensity of heartbeat sensations. We also conducted analyses with and without the inclusion of the guessing trial. Overall, it appeared that suicide attempters had lower heartbeat perception accuracy across all three trials, with the strongest differences occurring during the least confounded condition (i.e. the no-guess trial). We did not investigate cardiac interoception using a more rigorous and ecologically valid form of perturbation, such as double-blinded infusions of isoproterenol (*Khalsa et al., 2009*), but would expect that blunted interoception in suicide attempters in a similar context would constitute robust evidence replicating the present findings. Lastly, we should note that none of the interoceptive tasks applied in this study have demonstrated sufficient reliability to be considered appropriate for implementation in prognostic assessments of suicidality in clinical settings.

## Conclusion

We find that suicide attempters exhibit evidence of 'interoceptive numbing' characterized by increased tolerance for aversive respiratory and nociceptive sensations, reduced awareness of the heartbeat, and blunted activity in the dorsal mid and posterior insular cortex, a region of the brain associated with the primary representation of visceral afferent signals. The presence of these specific interoceptive deficits among individuals with prior suicide attempts reveals a possible role of interoceptive dysfunction in distinguishing individuals at risk of suicide.

## Materials and methods

### Participants

We performed a retrospective analysis from a pre-existing dataset containing the first 500 participants of the Tulsa-1000 (T-1000) cohort, a naturalistic longitudinal study of 1000 individuals with mood, anxiety, substance use, and/or eating disorders (*Victor et al., 2018*). Participants were considered eligible for T-1000 study entry if they fulfilled any of the following symptom criteria: Patient Health Questionnaire (PHQ-9; *Kroenke et al., 2001*)$\geq$10 and/or Overall Anxiety Severity and Impairment Scale (OASIS; *Campbell-Sills et al., 2009*)$\geq$8, and/or Drug Abuse Screening Test (DAST-10; *McCabe et al., 2006*) score >3, and/or Eating Disorder Screen (SCOFF; *Morgan et al., 2000*) score $\geq$2. Please refer to *Victor et al. (2018)* for a detailed description of the T-1000 inclusion criteria and study procedures. All participants provided written informed consent and received financial compensation for their involvement, and all procedures were approved by the Western Institutional Review Board.

Participants were included in the suicide attempter group (*n* = 34) if they endorsed making a suicide attempt at any point during the previous five years as documented in the Columbia Suicide Severity Rating Scale (CSSRS; *Posner et al., 2011*) and/or life-chart interviews (*Aupperle et al., 2020*), which were conducted during the baseline data collection period, and used to gather information about each participant's lifetime psychosocial, medical, educational, occupational, and treatment history (*Victor et al., 2018*). We used a propensity score matching algorithm for psychiatric reference sample identification (MatchIt package in R *De et al., 2011*, 1:2 nearest neighbor method without replacement), resulting in a group of non-attempter participants who denied having ever made a suicide attempt, and who exhibited similar screening symptoms on the PHQ-9, SCOFF, DAST, and OASIS scales (*Table 1*). To maximize the amount of data available for analysis, non-attempter participants (N = 239) were only matched to suicide attempters if their data had been manually checked and they had complete observations of the behavioral and psychophysiological variables examined. Further information regarding participant inclusion and exclusion criteria, matching procedures, and suicide attempt method are provided in Appendix 1.

## Procedures

For a detailed description of general study procedures, please see Appendix 1.

### Breath-hold challenge

Each participant completed two inspiratory breath-hold trials, providing a brief measure of sensitivity to respiratory perturbation (i.e. air hunger). Participants were seated in front of a computer screen, fitted with a respiration belt (Biopac Systems, Inc), provided with a nose clip, and a tube-like breathing apparatus was placed into the mouth. During normal breathing, concentrations of $O_2$ and carbon dioxide $CO_2$ were analyzed from exhaled air, providing a baseline measurement. Participants were then instructed to inhale maximally and, at the end of inhalation, to begin holding their breath for as long as they were able to tolerate. Trial duration was limited to 2 min, with a 2-min rest period between each trial; participants were instructed to stop and breathe if they reached the time limit, but they were not informed of how long the time limit would be beforehand. Participants were instructed to exhale into the breathing apparatus when they were no longer able to tolerate the breath-hold. Following each breath-hold, participants provided ratings of the task (i.e. respiratory sensation intensity, unpleasantness, and difficulty) as well as ratings of associated psychological experiences (i.e., stress, required effort, breathlessness, urge to breathe, breathlessness, sensations of suffocation, fear of suffocation) on a visual analogue scale (VAS) ranging from 0 ('Not at all') to 100 ('Extremely').

### Cold-pressor challenge

Participants immersed their dominant hand in a circulating pool of water cooled to six degrees Celsius. They were asked to keep their hand submerged for as long as they could tolerate. Maximum trial duration was limited to 2 min, although, as in the breath-hold task, this was not disclosed beforehand. Throughout the task, participants made continuous real-time pain intensity ratings with their dominant hand on a scale ranging from 0 ('No pain') to 100 ('Worst pain imaginable'). These ratings were used to calculate each participant's peak pain rating, as well as the time elapsed prior to the ratings of mild (25 out of 100), moderate (50 out of 100), and peak pain (100 out of 100 or the participant's maximum pain rating). Afterwards each participant provided ratings of task unpleasantness, difficulty, and stress on a visual analogue scale (VAS) ranging from 0 ('Not at all') to 100 ('Extremely').

### Heartbeat perception task

To assess cardiac interoception, participants performed three trials of a heartbeat tapping task. Participants were instructed to tap a key on a keyboard every time they felt their heartbeat, without taking their pulse. Each trial was 60 s in duration. In the first trial ('guess'), subjects were instructed to tap every time they felt their heartbeat without taking their pulse. Guessing was encouraged if they felt unsure. In the next trial ('no guess'), guessing was discouraged and participants were asked to tap only when they felt confident in feeling their heartbeat. In the final trial ('perturbation'), participants were instructed to inhale deeply, hold their breath, and tap along with their perceived

heartbeats while sustaining the breath-hold. The breath-hold perturbation was expected to amplify cardiac sensations and presumably increase heartbeat perception accuracy. Guessing was also discouraged in this trial. Heartbeat perception accuracy was calculated using a common accuracy metric (*Schandry, 1981*). Afterwards, participants provided VAS ratings ranging from 'Not at all' (0) to 'Extremely' (100) to indicate their perceived heartbeat intensity, confidence in their ability to accurately estimate their heartbeat, and their assessment of task-related difficulty.

## Interoceptive attention task during fMRI

The interoceptive attention task engages selective attention toward naturally-occurring interoceptive sensations in order to amplify activity in brain regions underlying interoceptive processing. We and others have previously demonstrated that this task is effective at mapping the neural basis of interoceptive attention in healthy individuals and those with, depression, substance use disorders, or eating disorders (*Avery et al., 2014*; *Kerr et al., 2016*; *Simmons et al., 2013a*; *Avery et al., 2017*; *Stewart et al., 2019*). The task consisted of two types of trials: interoceptive trials and exteroceptive trials. During the interoceptive trials, the words 'HEART' or 'STOMACH' were presented in a black font against a white background with each trial lasting 10 s. During the interoceptive trials, participants were instructed to focus on the sensations in their heart or stomach. Trials involving stomach interoception were not examined in the current study. Each interoceptive and exteroceptive stimulus was presented 12 times. During the exteroceptive trials, the word 'TARGET' was presented on the screen in black text against a white background. The color of the word periodically changed from black to various lighter shades of gray; throughout the duration of the 10 s trial, participants were instructed to focus on the intensity of the color change. To ensure that participants remained attentive during the task, following one-half of the trials, participants were asked to rate the intensity of the sensations from their heart or stomach or the intensity of the color change, on a scale from 0 to 6, with 0 indicating 'No sensation' (interoceptive) or 'No change in color' (exteroceptive) and six indicating an 'Extremely' intense sensation (interoceptive) or an 'Extremely' intense color change (exteroceptive). Participants performed this task over two scanning runs, each lasting 360 s.

## Analysis of demographic, behavioral, and physiological data

We conducted analyses of demographic, clinical, behavioral, and physiological data using the R base statistical software package version 3.5.1 (*R Development Core Team, 2013*). The 'TableOne' package (version 0.9.3; *Yoshida et al., 2019*) was used to display summaries of clinical characteristics between groups. LME analyses were conducted using the 'lmerTest' package version 3.1.1 (*Kuznetsova et al., 2017*). A marginal ANOVA was used on each LME model to examine *F*-tests for interactions and main effects. In the event of significant interactions, the summaries of LME fixed effects were examined to clarify which factors were driving the effect. The Kenward-Roger approximation of degrees of freedom was used for all LME analyses. R-squared estimates for fixed-effects were computed using the 'r2glmm' package in R (*Jaeger, 2017*) as described in *Edwards et al. (2008)*. Tables depicting model output were generated using the 'sjPlot' package (version 2.6.2; *Lüdecke, 2018*) and figures were created using the 'ggplot2' package (version 3.0.0; *Wickham, 2011*). VAS ratings for each task were also compared between groups. Since a proportion of the VAS ratings were not normally distributed, Mann-Whitney tests, which are robust to deviations from normality, were used to compare ratings between groups. Where applicable, a Benjamini-Hochberg correction was applied to minimize the false discovery rate (FDR) associated with repeated testing. We provide specific details for the analysis of each task below. The source code for our primary analyses and figures has also been provided.

## Breath-hold challenge

Three LME models were used to examine the relationship between group, trial repetition, the interaction between group and trial, and three outcome variables: breath-hold duration, post breath-hold $CO_2$ concentration, and post breath-hold $O_2$ concentration. For each model, group, timepoint, and their interaction were included as fixed effects. A participant identifier was included as a random effect, and hold duration, post-hold $CO_2$ concentration, or post-hold $O_2$ concentration were specified as dependent measures. A Benjamini-Hochberg correction was applied across these three models. Additionally, Benjamini-Hochberg corrected Mann-Whitney tests were used to examine group

differences in VAS ratings of intensity, unpleasantness, difficulty, stress, required effort, breathlessness, urge to breathe, breathlessness, sensations of suffocation, and fear of suffocation following the breath-hold tasks.

### Cold-pressor challenge

We used a LME model to examine the relationship between group and the amount of time (seconds) elapsed from the start of the cold-pressor challenge until the participant reached four markers of pain intensity: mild pain, moderate pain, peak pain, and task discontinuation (i.e. hand removal from the water). Group, timepoint, and the interaction between group and timepoint were included as fixed effects. A participant identifier was included as a random effect, and duration in seconds was the specified dependent measure. VAS ratings of pain intensity, unpleasantness, difficulty, and stress were also compared between groups using Benjamini-Hochberg corrected Mann-Whitney tests.

### Heartbeat perception task

We used a LME model to examine the relationship between group and heartbeat tapping accuracy across the three task conditions (i.e. guess, no-guess and breath-hold). Heartbeat tapping accuracy was included as the dependent measure, the interaction between group and condition was modeled as a fixed effect, and a participant identifier was specified as a random effect. Additionally, based on a recent study indicating that heartbeat perception accuracy scores are potentially confounded by guessing (*Desmedt et al., 2018*), a post-hoc decision was made to apply a second model that omitted the guessing score from the analysis, focusing only on the no-guess and breath-hold perturbation trials. Benjamini-Hochberg corrected Mann-Whitney tests were used to examine group differences in mean VAS ratings of heartbeat perception confidence, task difficulty, and heartbeat intensity across the no-guess and breath-hold trials.

## Analysis of interoceptive attention task during fMRI

### Data acquisition and imaging parameters

Structural and functional magnetic resonance images were acquired using a General Electric (GE) Discovery MR750 3 Tesla MRI scanner. A 3D MPRAGE sequence obtained high-resolution anatomical images (FOV = 240 mm x 192 mm, slices/volume (axial) = 186, slice thickness = 0.9 mm, image matrix = 256×256, voxel volume = 0.938×0.938 × 0.9 mm, TR/TE = 5/2.012 ms, acceleration factor R = 2, flip angle = 8°, inversion/delay time TI/TD = 725/1400 ms, scan time = 340377 ms) using an 8-channel receive-only head coil (GE). Functional data were collected as echo-planar image (EPI) volumes depicting BOLD contrast (180 EPI volumes per run, slice thickness = 2.9 mm, voxel volume = 1.875×1.875 × 2.9 mm, acquisition matrix = 96×96, TR = 2000 ms, TE = 27 ms, flip angle = 78°, axial-oblique slices, 39 slices per volume, scan time = 360 s) using an eight-channel head array coil (GE), with a sensitivity encoding (SENSE) factor of 2 to minimize EPI distortions while also increasing the number of slices collected per TR.

### Preprocessing, and subject-level analysis

Data preprocessing was conducted using afni_proc.py (*Cox, 1996*). The first three volumes of the functional scans were discarded to allow the signal to reach T1 equilibrium, and a despiking algorithm was used to remove any transient signal spikes from the data. For each participant, the remaining volumes were corrected for differences in slice acquisition time; head motion was corrected by rigid body translation and rotation; the first volume of the functional run (before discarding three volumes) was coregistered to the anatomical coordinates of the participant's structural scan by linear warping, then normalized to the Montreal Neurological Institute (MNI) template and resampled to 2 × 2 × 2 mm$^3$ voxels. The EPI data were then smoothed using a 4 mm full-width at half-maximum Gaussian kernel, and the value for each EPI volume was normalized to percent signal change using each voxel's average signal across the time course.

The imaging data were analyzed at the subject level using a multiple linear regression model, with regressors for each task condition (i.e. cardiac attention, stomach attention, exteroceptive attention, and response periods). To adjust the model for the shape and delay of the BOLD function, task regressors were constructed by convolution of a block function having a 5- or 10 s width (depending on the trial duration) beginning at the onset of occurrence of each condition. Nuisance

regressors included each run mean, linear, quadratic, and cubic signal trends, as well as six head motion variables (three translations, three rotations).

## Group analysis

We identified the insular cortex as an a priori region of interest due to its well-documented involvement in interoceptive attention (*Avery et al., 2014*; *Kerr et al., 2016*; *Simmons et al., 2013a*; *Avery et al., 2017*; *Stewart et al., 2019*), and focused our analysis on cardiac interoception. A mask of the left and right insula was defined using the N27 anatomical atlas within AFNI. The AFNI program 3dttest++ was used to examine group differences in interoceptive attention. For the interoceptive attention condition, all subject-level beta coefficients represented signal change relative to the exteroceptive condition. A small volume correction of $p < 0.005$ was applied within the insula, and a voxel-wise threshold of $p < 0.0005$ was set for the rest of the brain. Results were then corrected using a cluster-size threshold of $\alpha < 0.05$. To accurately estimate the cluster sizes necessary to achieve familywise error correction, we applied the –Clustsim correction in 3dttest++, which employs randomization and permutation simulation to produce cluster-level threshold values that adequately control the false positive rate (*Cox et al., 2017*).

# Acknowledgements

The authors thank W Kyle Simmons, PhD, for helpful discussions that motivated the initial work on this project, Rachel Lapidus, MA, for helpful comments offered on the manuscript, Maria Puhl, PhD, and Wesley Thompson, PhD, for providing statistical consultation, the Tulsa-1000 clinical assessment team for participant recruitment and data collection, and the MRI technologists at the Laureate Institute for Brain Research for MRI data acquisition. The authors acknowledge Austin Lignieres, BS, James Hale, and Max Paulus for assisting with physiological data inspection and correction. This work has been supported in part by The William K Warren Foundation, by NIH/National Institute of Mental Health grant K23MH112949 (to SSK), and the National Institute of General Medical Sciences Center Grant Award Number 1P20GM121312 (JLS, MPP, SSK). The content is solely the responsibility of the authors and does not necessarily represent the official views of the National Institutes of Health. Results presented in part at the Anxiety and Depression Association of America 39th Annual Scientific Meeting, March 2019, and at the Society of Biological Psychiatry 74th Annual Meeting, May 2019, in Chicago, IL.

The ClinicalTrials.gov identifier for the clinical protocol associated with data published in the current paper is NCT02450240, 'Latent Structure of Multi-level Assessments and Predictors of Outcomes in Psychiatric Disorders'.

The Tulsa 1000 Investigators include the following contributors: *Robin L Aupperle, Ph.D., Jerzy Bodurka, Ph.D., Justin S Feinstein, Ph.D., Sahib S Khalsa, M.D., Ph.D., Rayus Kuplicki, Ph.D., Martin P Paulus, M.D., Jonathan Savitz, Ph.D., Jennifer L Stewart, Ph.D., and Teresa A Victor, Ph.D.*

# Additional information

### Group author details

**Tulsa 1000 Investigators**

**Robin L Aupperle**: Laureate Institute for Brain Research, Tulsa, United States; **Jerzy Bodurka**: Laureate Institute for Brain Research, Tulsa, United States; **Yoon-Hee Cha**: Laureate Institute for Brain Research, Tulsa, United States; **Justin Feinstein**: Laureate Institute for Brain Research, Tulsa, United States; **Jonathan B Savitz**: Laureate Institute for Brain Research, Tulsa, United States; **Teresa A Victor**: Laureate Institute for Brain Research, Tulsa, United States

### Funding

| Funder | Grant reference number | Author |
| --- | --- | --- |
| William K. Warren Foundation | | Martin P Paulus |
| National Institute of Mental Health | K23MH112949 | Sahib S Khalsa |

| National Institute of General Medical Sciences | P20GM121312 | Jennifer L Stewart Martin P Paulus Sahib S Khalsa |

The funders had no role in study design, data collection and interpretation, or the decision to submit the work for publication.

## Author contributions

Danielle C DeVille, Conceptualization, Formal analysis, Investigation, Methodology; Rayus Kuplicki, Formal analysis, Methodology; Jennifer L Stewart, Tulsa 1000 Investigators, Writing - review and editing, Conceptualization, Supervision, Funding acquisition, Methodology; Martin P Paulus, Sahib S Khalsa, Conceptualization, Supervision, Funding acquisition, Methodology

## Author ORCIDs

Danielle C DeVille (iD) https://orcid.org/0000-0001-8208-2705
Sahib S Khalsa (iD) https://orcid.org/0000-0003-2124-8585

## Ethics

Human subjects: All study procedures were approved by the Western Institutional Review Board (WIRB protocol #20142082) and all research participants provided written informed consent prior to participation in the research.

## Decision letter and Author response

Decision letter https://doi.org/10.7554/eLife.51593.sa1
Author response https://doi.org/10.7554/eLife.51593.sa2

## Additional files

### Supplementary files

• Source code 1. R code for primary analyses and creation of *Figures 1–4*.

• Supplementary file 1. Output of linear mixed effects models for breath hold duration, $CO_2$, and $O_2$.

• Supplementary file 2. Output of linear mixed effects model for cold pressor challenge.

• Supplementary file 3. Output of linear mixed effects model for heartbeat perception models.

• Transparent reporting form

### Data availability

We are unable to publicly release our complete raw dataset due to the fact that our research engaged human participants to examine a very sensitive topic (history of suicide attempt), and the fact that our dataset contains information that could potentially identify individuals. However, should another investigator wish to obtain access to other aspects of our data for the purposes of re-analysis or verification of our findings, we would be willing to facilitate this process upon request, provided that the confidentiality of the participants can be protected. Individuals who are interested in obtaining access to this data should contact the corresponding author. De-identified processed source data and R code corresponding to each figure has been uploaded.

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

## Appendix 1

### General procedures

The data used in the current study were collected over three to four visits to the Laureate Institute for Brain Research facility. The initial visits involved completion of the following clinical assessment measures: Patient Health Questionnaire (PHQ-9), Overall Anxiety Severity and Impairment Scale (OASIS), Drug Abuse Screening Test (DAST), Eating Disorder Screen (SCOFF), an assessment of medication status and medical history, a demographics questionnaire, the Mini International Neuropsychiatric Interview (*Sheehan et al., 1998*), and the Columbia Suicide Severity Rating Scale (C-SSRS)(*Posner et al., 2011*). There were no differences in diagnosis or medication status between groups, as illustrated in *Appendix 1— tables 1–2*. During the initial visit, participants completed an additional battery of self-report scales, from which the Multidimensional Assessment of Interoceptive Awareness (*Mehling et al., 2012*), the Anxiety Sensitivity Index (*Taylor et al., 2007*), and the Toronto Alexithymia Scale *Bagby et al., 1994*) were selected and included in current analyses (see *Appendix 1—table 3*; no significant group differences were observed on these measures after correcting for multiple comparisons). The latter visits involved completion of a fMRI scan and, on a separate date, completion of psychophysiological tasks including the breath-hold challenge, cold pressor challenge, and heartbeat perception tasks. An experimenter remained in the room with each participant but out of sight during the psychophysiological tasks, and video recordings of each participant were obtained to ensure compliance with instructions.

**Appendix 1—table 1.** Percent of individuals meeting diagnostic criteria for psychiatric disorders per the Mini International Neuropsychiatric Interview (MINI) at the time of data collection.

|  | Suicide attempters (*n* = 34) | Non-attempters (*n* = 68) |
|---|---|---|
| Major depressive disorder | 74% (n = 25) | 74% (n = 50) |
| Anxiety disorders | 62% (n = 21) | 65% (n = 44) |
| Post-traumatic stress disorder | 32% (n = 11) | 28% (n = 19) |
| Substance use disorder (excluding alcohol) | 35% (n = 11) | 41% (n = 28) |
| Alcohol use disorder | 26% (n = 9) | 24% (n = 16) |
| Eating disorders | 9% (n = 3) | 3% (n = 2) |
| Mean (SD) number of diagnoses per individual | 3.2 (1.9) | 3.1 (1.9) |

Note: There were no significant differences in the proportion of diagnoses between suicide attempter and non-attempter groups. 'Anxiety disorder' includes any of the following: social anxiety disorder, general anxiety disorder, and/or panic disorder with or without agoraphobia. 'Eating disorder' includes a diagnosis of anorexia nervosa, bulimia nervosa, or binge eating disorder. Total percentages will not add up to 100% as several patients met criteria for more than one disorder.

**Appendix 1—table 2.** Percent of individuals taking psychotropic medications by drug class.

|  | Suicide attempters (*n* = 34) | Non-attempters (*n* = 68) |
|---|---|---|
| SSRIs | 38% (n = 13) | 29% (n = 20) |
| SNRIs | 14.7% (n = 5) | 7.4% (n = 5) |
| Bupropion | 17.6% (n = 6) | 8.8% (n = 6) |
| Tricyclic Antidepressants | 8.8% (n = 3) | 0 |
| Tetracyclic Antidepressants | 14.7% (n = 5) | 1.5% (n = 1) |

*Appendix 1—table 2 continued on next page*

*Appendix 1—table 2 continued*

|  | Suicide attempters (*n* = 34) | Non-attempters (*n* = 68) |
|---|---|---|
| Atypical antipsychotics | 23.5% (n = 8) | 7.4% (n = 5) |
| Anticonvulsants | 26.5% (n = 9) | 10.3% (n = 7) |
| Benzodiazepines | 26.5% (n = 9) | 13.2% (n = 9) |
| Stimulants | 5.8% (n = 2) | 13% (n = 9) |
| Narcotics (e.g., Suboxone) | 11.8% (n = 4) | 11.8% (n = 8) |
| Opioid Antagonists | 5.8% (n = 2) | 1.5% (n = 1) |
| Lithium | 5.8% (n = 2) | 0 |

Note: 'SSRI'=selective serotonin reuptake inhibitor; 'SNRI'=serotonin and norepinephrine reuptake inhibitor

**Appendix 1—table 3.** Self-report measures by group.

|  | Attempter (*n* = 34) | Non-attempter (*n* = 68) |  |  |
|---|---|---|---|---|
|  | Mean (SD) |  | *p* | Adjusted *p* |
| *Multidimensional Assessment of Interoceptive Awareness (MAIA)* |  |  |  |  |
| Attention Regulation | 2.7 (1.2) | 2.9 (1.0) | 0.23 | 0.37 |
| Body Listening | 2.0 (1.3) | 2.4 (1.1) | 0.10 | 0.37 |
| Emotional Awareness | 3.4 (1.2) | 3.7 (0.9) | 0.21 | 0.37 |
| Not Distracting | 1.8 (0.9) | 1.5 (0.9) | 0.23 | 0.37 |
| Noticing | 3.5 (0.8) | 3.6 (1.0) | 0.76 | 0.87 |
| Not Worrying | 2.7 (1.0) | 2.5 (1.1) | 0.44 | 0.59 |
| Self-Regulation | 2.4 (1.2) | 2.8 (1.1) | 0.90 | 0.90 |
| Body Trust | 2.7 (1.3) | 3.2 (1.0) | 0.02 | 0.16 |
| *Anxiety Sensitivity Index (ASI)* |  |  |  |  |
| Cognitive Concerns | 6.9 (6.7) | 6.1 (6.7) | 0.52 | 0.88 |
| Physical Concerns | 5.8 (5.7) | 6.3 (5.7) | 0.65 | 0.88 |
| Social Concerns | 11.6 (6.8) | 11.4 (6.8) | 0.88 | 0.88 |
| Total ASI | 24.3 (17.1) | 23.8 (12.9) | 0.86 | 0.88 |
| *Toronto Alexithymia Index (TAS-20)* |  |  |  |  |
| Difficulty Identifying Feelings | 18.8 (6.5) | 18.1 (5.0) | 0.54 | 0.72 |
| Difficulty Describing Feelings | 14.7 (3.2) | 14.7 (3.2) | 0.65 | 0.72 |
| Externally Oriented Thinking | 25.3 (3.2) | 26.4 (3.0) | 0.09 | 0.36 |
| Total TAS | 58.8 (10.2) | 59.5 (8.2) | 0.72 | 0.72 |

Note: The adjusted p-values are corrected using the Benjamini-Hochberg adjustment.

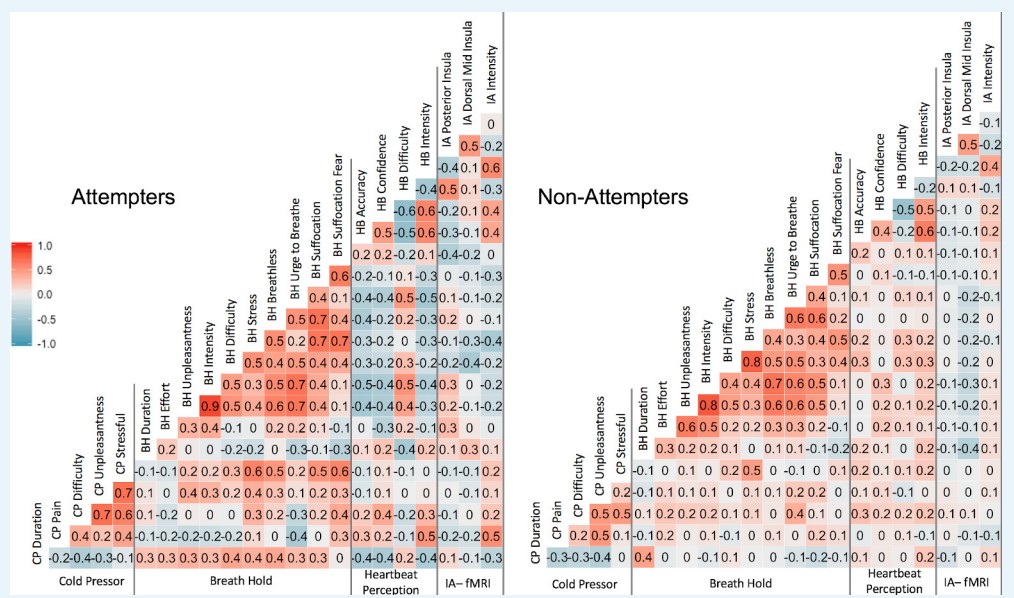

**Appendix 1—figure 1.** Pearson correlation coefficients across measures. 'CP' = cold-pressor, 'BH' = breath-hold (computed using participants' mean duration and VAS ratings across trial 1 and trial 2, 'HB' = heartbeat perception task (computed using the mean of the no-guess and perturbation conditions). IA-fMRI = interoceptive attention to heartbeat sensations during fMRI.

# Matching procedures and participant selection

Several individuals from the first 500 participants of the Tulsa 1000 dataset were not included in the current analysis. These included: 1) participants reporting suicide attempts occurring more than 5 years prior to entry into the study ($n = 51$), 2) participants with missing or incomplete C-SSRS data ($n = 44$), 3) participants who reported a history of *aborted* suicide attempts only ($n = 7$), defined as having made a suicide plan with intent but never taking any action towards it, and 4) participants who were initially categorized as non-attempters but made subsequent attempts after data collection ($n = 3$). Our decision to only include suicide attempters who had made attempts within the past 5 years was implemented in response to literature finding that individuals who have made suicide attempts in the previous 5 years exhibit more self-reported interoceptive deficits than those who made attempts more than 5 years ago (**Forrest et al., 2015**). Following the matching procedure, there were no statistically or clinically significant differences between non-attempters and attempters on the PHQ-9, DAST, SCOFF, or OASIS, with each group's mean on these measures matched within < 1 point of the other group. While a group of psychiatrically healthy individuals participated in the T1000 study, they were not incorporated into the current study.

# Missing data values

The data from 2 to 5 suicide attempters were not eligible for inclusion in each of the breath-hold analyses, either due to missing or unusable physiological data. For the breath hold duration analysis, the sample was reduced to 32 participants in the suicide attempter group. All 68 participants in the non-attempter group had usable data for calculation of breath-hold duration. In the analyses comparing oxygen ($O_2$) and carbon dioxide ($CO_2$) concentrations following the breath-hold, there were 29 suicide attempters with complete post breath-hold measures of $O_2$, and 28 with complete post breath-hold measures of $CO_2$. There were seven individuals from the suicide attempter group with missing cold-pressor data; as such, the analyses reported here reflect a comparison of 27 attempter and 68 non-attempter participants. There were two suicide attempters with missing data from the heartbeat

perception task. Additionally, a total of 7 participants (four non-attempters and two attempters) were not included in the fMRI analysis, for the following reasons: two participants were not scanned, one participant had significant slice artifacts, three participants had poor anatomical-functional image alignment, and one participant had an average head motion greater than 0.3 millimeters.

## Suicide attempt methods and recency of attempts

Within the attempter group, 26% ($n = 9$) attempted suicide via asphyxiation, 11% ($n = 3$) by injury involving cutting the wrists or other areas, 15% ($n = 5$) by other injuries, such as jumping into traffic, intentional car accidents, or attempting to fire a gun, and 50% ($n = 17$) by self-poisoning (e.g., overdose). A total of 62% ($n = 24$) of individuals in the attempter group endorsed one previous suicide attempt, 24% ($n = 8$) reported two previous attempts, 6% ($n = 2$) reported three previous attempts, and 9% ($n = 3$) reported four or more previous suicide attempts. For participants with multiple attempts, the attempt method reported here reflects their most recent attempt. Of all attempter participants identified, a total of 11 had identified as having made their most recent suicide attempt within 1 year of data collection, six reported making a suicide attempt within 2 years of data collection, and 11 participants reported attempting suicide between 2 and 5 years prior to data collection.

## Appendix 2

### Assessing effects of psychotropic medication status

To address the potential concern that our results might be attributed to the effects of psychotropic medication rather than suicide attempt history, we re-examined our primary outcome variables after covarying for medication status.

#### Heartbeat perception accuracy

To examine whether the observed group difference between suicide attempters' and non-attempters on heartbeat tapping accuracy remained after covarying for the effects of psychotropic medication status, we included psychotropic medication status (i.e., medicated or unmedicated) as a fixed effect covariate in our linear mixed effects (LME) model and re-analyzed the model, again using analysis of variance (ANOVA). After adjusting the model for medication, the main effect of suicide group (attempter vs. non-attempter) remained significant ($F(1,140.32) = 9.3$, p = 0.003) as did the significant interaction between group and condition ($F(1,97.72) = 4.37$, $p = 0.039$). There was no significant effect of medication status ($F(1,96.7) = 0.74$, p = 0.392).

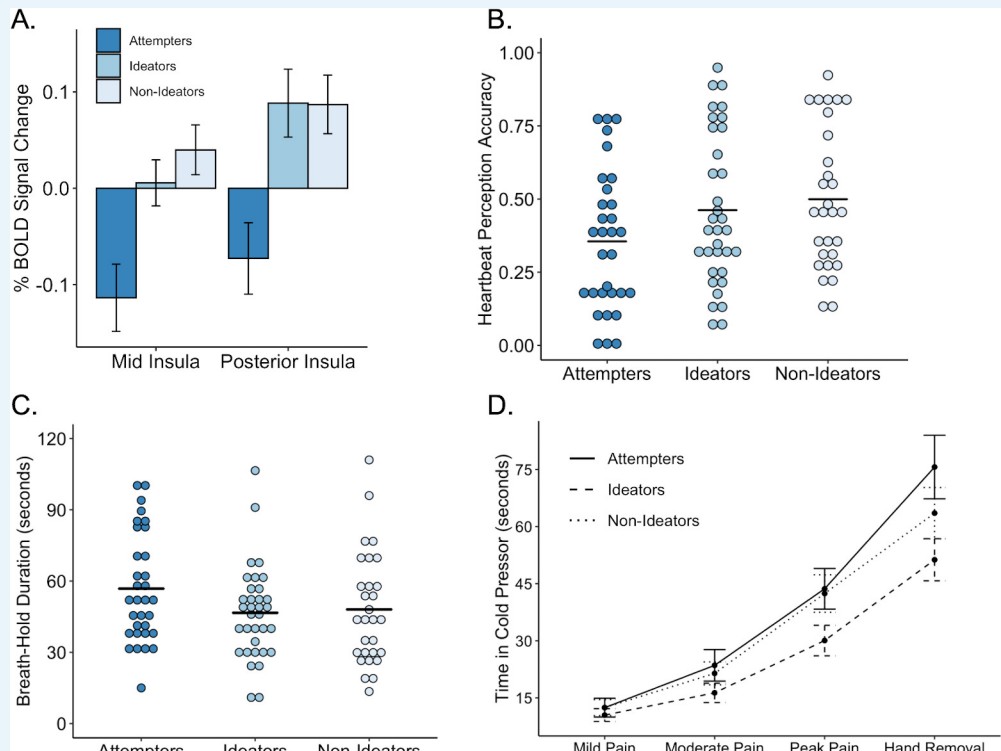

**Appendix 2—figure 1.** Performance on interoceptive measures across three groups: suicide attempters, suicide ideators, and non-ideators. (**A**) During attention to interoceptive sensations (vs. exteroceptive sensations) suicide attempters exhibited lower blood oxygen level-dependent (BOLD) signal in the right dorsal mid insula compared to ideators (p = 0.004) and non-ideators (p < 0.001). Suicide attempters also exhibited lower BOLD activation in the posterior insula relative to ideators (p = 0.001) and non-ideators (p = 0.002) during interoceptive attention. (**B**) Suicide attempters exhibited lower levels of heartbeat perception accuracy relative to that of ideators (p = 0.012) and non-ideators (p = 0.007) during the no-guess and perturbation trials of the heartbeat perception task. The mean of the no-guess and perturbation trials is illustrated. (**C**) There was no significant difference between attempters

and ideators (p = 0.062), nor between attempters and non-ideators (p = 0.110) in breath hold duration across trials. (**D**) Relative to ideators, suicide attempters sustained the cold pressor longer after reaching peak pain (p = 0.001); no significant differences were observed between attempters and non-ideators (p = 0.074). Error bars indicate + / - 1 standard error.

## Blunted insula signal during interoceptive attention

To examine whether the observed group difference between suicide attempters' and non-attempters' signal in the mid and posterior insula during cardiac attention could be attributed to psychotropic medication status, we extracted the subject-level beta values from each insula cluster corresponding to the contrast between the cardiac and exteroceptive attention. We then built a linear model with the subject-level beta values as the dependent measure, suicide attempt history (attempter vs. non-attempter) and medication status as predictor variables. We did not use LME for this analysis as no repeated measures were involved. The effect of suicide attempt history was still significant in both regions (mid insula: $F_{(1,92)} = 13.63$, p = 0.0004; posterior insula: $F_{(1,92)} = 12.62$, p = 0.0006). Moreover, there was no effect of medication status on the right mid insula or posterior insula beta values during cardiac (vs. exteroceptive) attention (mid insula: $F_{(1,92)} = 2.36$, p = 0.13; posterior insula: $F_{(1,92)} = 3.22$, p = 0.08).

## Breath-hold task

To assess the role of medication status, we included psychotropic medication status as a covariate in the LME models used to assess for group differences in breath hold task performance. Specifically, three models were built to examine the relationship between suicide attempt history, breath hold trial, and the following outcome variables 1) breath-hold duration, 2) post breath-hold expired oxygen concentrations, and 3) post breath-hold expired carbon dioxide concentrations, after covarying for medication status. For the model examining breath-hold duration, medication status was a non-significant predictor of duration ($F_{(1,96.8)} = 2.62$, p = 0.109). However, the group difference in breath-hold was no longer statistically significant when medication status was included as a covariate ($F_{(1,119.38)} = 2.8$, p = 0.095). For the models examining oxygen and carbon dioxide concentrations, the group differences between suicide attempters' and non-attempters' post breath-hold oxygen and carbon dioxide concentrations remained statistically significant even after covarying for medication status, (*oxygen: $F_{(1,91.08)} = 4.13$ p = 0.044; carbon dioxide: $F_{(1,91.04)} = 5.55$*, p = 0.020). No effects of medication status on post breath-hold concentrations of oxygen ($F_{(1,94.12)} = 0.266$, p = 0.61) or carbon dioxide ($F_{(1,93.84)} = 0.127$, p = 0.722) were observed.

## Cold-pressor challenge

To examine whether suicide attempters exhibited an increased tolerance for cold-pressor pain after covarying for the effects of medication status, we re-analyzed our LME models with medication status included as a covariate. After adjusting the model for the effects of medication status, the interaction between time and suicide attempt group remained statistically significant ($F_{(3,277.36)} = 2.9$, p = 0.036). Examination of fixed effects revealed that, while suicide attempters exhibited no differences in the amount of time elapsed prior to their ratings of mild, moderate, or peak pain, they kept their hands submerged in the water for a longer period of time ($t_{(278.56)} = 2.79$, p = 0.006). There was no effect of medication status ($F_{(1,91.37)} = 0.039$, p = 0.84).

## Discussion of psychotropic medication status

After re-examining the data to evaluate a potential influence of differences in psychotropic medication status between suicide attempters and non-attempters, we found a similar overall pattern of results. Specifically, across all tasks we continued to see evidence suggestive of 'interoceptive numbing' characterized by a heightened tolerance for aversive interoceptive

sensations (pain tolerance via the cold-pressor challenge, elevated carbon dioxide during breath hold) and reduced awareness of non-aversive sensations (decreased heartbeat perception accuracy and reduced activation of posterior insula during interoceptive attention to heartbeat sensations). These findings further reinforce the notion that blunted awareness of interoceptive sensations and attenuated responses to bodily threat may distinguish individuals at risk for suicide.

## Assessing for potential effects of suicidal ideation

In the current study we evaluated evidence for interoceptive processing differences between individuals with a history of suicide attempts within the last 5 years and individuals with no suicide attempt history. We did not specifically address the role of suicidal ideation. Our primary rationale for not focusing on ideation history in the current study was twofold: 1) suicide attempt history has a stronger theoretical link to blunted interoceptive processing than suicidal ideation, as individuals who have made a prior suicide attempt have demonstrated an ability to override homeostatic signals through their suicidal actions; and 2) we had a limited assessment of suicide ideation history as the scale we used to assess this, the Columbia Suicide Severity Rating Scale (C-SSRS), accounts only for the worst lifetime presence of ideation and does not prompt the interviewer to collect dates of ideation. Recognizing that readers may seek more information regarding suicidal ideation within our sample, we sought to determine whether the observed relationship between interoception and suicidal behavior could be simply attributed to differences in ideation. We addressed this to the best of our ability by 1) re-analyzing our four primary findings after including the intensity of suicidal ideation as a covariate, and 2) dividing our non-attempter group on the basis of lifetime ideation and examining differences between suicide attempters, lifetime ideators without suicide attempts, and non-ideators. We report these analyses below.

Approximately 54% of non-attempters in our sample ($n = 34$) endorsed a lifetime history of suicidal ideation. Individuals completing the C-SSRS are also graded with an 'Intensity of Ideation' (IOI) score, which is calculated from the participants' responses to items regarding the frequency, duration, controllability, deterrents, and reasons associated with their suicidal ideation. On the IOI scale, a score of 0 is obtained for individuals with no ideation, whereas ideators have a possible score ranging from 2 to 25. In our sample, the mean IOI score obtained from non-attempters was approximately 5.2 (SD = 7.1), whereas the suicide attempter group had a mean IOI of 15.2 (SD = 7.0); the difference in IOI between groups was statistically significant ($t(87) = -6.04$, $p < 0.001$).

### Blunted insula signal during interoceptive attention

Using the subject-level beta coefficient values from the insula clusters corresponding to the contrast between cardiac and exteroceptive attention, we constructed a linear model to examine the relationship between suicide attempt status and insula BOLD signal in each cluster after covarying for each participant's IOI score. Examination of this model revealed that IOI was not significantly associated with BOLD signal differences in the dorsal mid insula ($t(81) = -5.82$, $p = 0.018$, $\beta = -0.07$) or the posterior insula ($t(81) = -1.31$, $p = 0.18$, $\beta = -0.16$) and that after including IOI in the models, attempt history remained significant (*Mid Insula: $t(81) = -2.80$, $p = 0.006$, $\beta = -0.34$; Posterior Insula: $t(81) = -2.74$, $p = 0.008$, $\beta = -0.32$*).

To examine differences between attempters', ideators', and non-ideators' signal in the insula during cardiac attention, we conducted a linear model using the subject-level betas in each cluster as the outcome measure and group (i.e., attempter, ideator, or non-ideator). For the right dorsal mid-insula, the attempters' BOLD signal was significantly lower relative to that of ideators and non-ideators (*attempters vs. ideators: $t(88) = 2.98$, $p = 0.004$, $\beta = 0.34$; attempters vs. non-ideators: $t(88) = 3.72$, $p < 0.0001$, $\beta = 0.42$*). The findings were similar for the posterior insula cluster, such that the attempters' exhibited significantly lower BOLD signal relative to that of ideators ($t(88) = 3.35$, $p = 0.001$, $\beta = 0.38$) and non-ideators ($t(88) = 3.21$, $p = 0.002$, $\beta = 0.36$).

## Heartbeat perception accuracy

We re-analyzed the linear mixed effects model to examine differences in heartbeat perception accuracy during the no-guess and breath-hold conditions between attempters and non-attempters after covarying for the effects of IOI. Examination of fixed effects revealed that when IOI and suicide attempt history were considered together, IOI was non-significant (t (85.23) = −0.011, p = 0.991, β = 0.00). After covarying for IOI, the effect of attempt history remained statistically significant, (t(115.1) = −2.45, p = 0.016, β = −0.30).

After dividing the non-attempter group on the basis of lifetime ideation, a linear mixed model was constructed to examine differences between attempters, lifetime ideators, and non-ideators. Examination of fixed effects revealed that both ideators and non-ideators were more accurate than attempters (*ideators vs. attempters:* t(138.14) = 2.54, p = 0.012, β = 0.25; *non-Ideators vs. attempters:* t(138.14) = 2.78, p = 0.007, β = 0. 32).

## Breath-hold challenge duration

A linear mixed effects model was used to examine the effect of attempt history on overall breath hold duration after covarying for IOI. Examination of fixed effects in the model containing IOI and attempt history revealed no significant effect of IOI (t(83.8) = −1.31, p = 0.195, β = −0.16). With IOI included as a covariate, the effect of attempt history remained significant (t(96.6) = 2.32, p = 0.022, β = 0.29).

After dividing the non-attempters on the basis of lifetime ideation, a LME model was used to examine differences in breath hold duration between attempters, lifetime ideators, and non-ideators. It was observed that attempters sustained the breath hold for approximately 11 s longer than ideators, and 10 s longer than non-ideators. However, the group differences were no longer statistically significant (*attempters vs. ideators:* t(113.18) = −1.88, p = 0.062; *attempters vs. non-ideators*: t(113.12) = −1.61, p = 0.109).

## Cold-pressor challenge duration

We re-analyzed the LME model to examine the effect of attempt history on cold-pressor challenge duration after covarying for IOI. Examination of fixed effects revealed a significant effect of IOI (t(78.48) = −2.36, p = 0.02, β = −0.18). When IOI was included as a covariate, the interaction between attempt history and timepoint remained significant (t(239) = 2.19, p = 0.030, β = 0.12).

After dividing the non-attempters on the basis of lifetime ideation, a linear mixed effects model was used to examine the relationship between cold-pressor challenge duration and group among attempters, lifetime ideators, and non-ideators. It was noted that suicide attempters sustained the cold-pressor challenge for approximately 24 s longer than ideators, and 12 s longer than non-ideators. Examination of fixed effects revealed that attempters sustained the significantly longer than ideators after reaching peak pain (t(259.9)=−3.39, p < 0.001, β = −0.21), but not significantly longer than non-ideators (t(92) = −1.80, p = 0.08. β = −0.11) after reaching peak pain.

## Discussion of suicidal ideation

After re-examining the data to evaluate the influence of suicidal ideation, the overall pattern of results suggests that our primary findings are likely attributed to suicidal behavior, not the effects of suicidal ideation. When intensity of suicidal ideation was included as a covariate, the group differences we previously reported were unchanged across all tasks. Additionally, when we analyzed differences between attempters, ideators, and non-ideators, we found that that the suicide attempters, relative to ideators and non-ideators, exhibited heightened pain tolerance in the cold-pressor challenge, reduced heartbeat perception accuracy, and reduced activation of the mid and posterior insula during interoceptive attention (*Appendix 2—figure 1*). Although further research on the distinction between ideation and attempts is warranted,

our initial findings support the notion that the interoceptive deficits observed among suicide attempters are not explained by the presence of suicidal ideation alone.

