## [Decision Letter]

**Acceptance summary:**

The authors examine whether individuals with a history of suicide attempt differ in interoceptive processing, relative to a matched psychiatric reference sample. The study includes a battery of interoceptive tasks (breath-hold, cold-pressor, heartbeat perception, and fMRI). The results indicate that suicide attempters show increased tolerance for aversive respiratory/nociceptive signals, reduced cardiac interoceptive accuracy, and a decreased insula signal during interoception. The findings suggest that blunted awareness of interoceptive sensations and attenuated responses to bodily threat may help to distinguish individuals at risk of suicide. The reviewers and I are enthusiastic about the work.

**Decision letter after peer review:**

Thank you for submitting your article "Diminished responses to bodily threat and blunted cardiac interoception in suicide attempters" for consideration by *eLife*. Your article has been reviewed by two peer reviewers, and the evaluation has been overseen by a Reviewing Editor (Alexander Shackman) and a Senior Editor (Christian Büchel). The following individual involved in review of your submission has agreed to reveal their identity: Nils Kroemer (Reviewer #2).

The reviewers have discussed the reviews with one another and the Reviewing Editor has drafted this decision to help you prepare a revised submission.

The two reviewers were enthusiastic about the work:

• I think this is a very strong and important paper. Its key strength is the convergence of multiple methods. This helps address critiques that might be levelled at individual measures. I am impressed by how well the groups are matched in terms of key variables. Some would view differences in psychotropic medication between the groups to be a major flaw, but I think the approach the authors have taken is sensible. Overall, this paper is well guided by theory, methodologically strong, and potentially important clinically.

• This is a very important topic that is notoriously difficult to investigate well so I would like to commend the authors. This is arguably one of the best studies on this topic that I am aware of.

Essential Revisions:

1) Stats.

a) The description of the statistical analysis in the manuscript is currently not sufficient. I was happy to look at the code, but it should not be necessary to know R to understand what exactly the authors did. Therefore, more details on the models should be provided in the Materials and methods or the supporting information.

b) Relatedly, it was not obvious to me why the authors used the lme function for some analyses but not others (although it was even passed to a _gls variable) and sometimes the summary function or the anova function was used, not both.

c) In the manuscript, it was not clear to me which results were reported. Thus, it would be preferable to be a bit more precise in terms of the statistical descriptions and to include the output of the estimated models in the supporting information.

2) Cold pressor challenge.

a) The authors write: "Examination of fixed effects revealed that, while there were no significant group differences in the amount of time elapsed prior to reaching mild, moderate, and peak pain, suicide attempters kept their hands submerged in the cold water for significantly longer (18 seconds on average, SD = 25) than non-attempters after reaching peak pain (t(276)=2.8, p = 0.006, Cohen's d = .34)."

b) First, does the average refer to the group difference in duration and the SD to the pooled SD across groups?

c) Second, if it only refers to this category, why are the degrees of freedom so high that it appears as if a fixed effect across all repeated measurements was taken for the comparison?

d) Third, how was Cohen's d calculated because the study would not be large enough to find significant group differences of this magnitude if the degrees of freedom are calculated correctly (1-β = .36 for d = .34, n1 = 34, n2 = 68). This indicates that a formula was used to estimate d from t, which is only valid for non-nested t-contrasts ("simple" two-sample t-tests).

3) Relations amongst measures.

a) At the moment, DeVille et al. are not fully capitalizing on the strength of the design provided by the complementary measures of interoceptive awareness because the correlations between the interoceptive measures are not reported.

b) The authors report associations between percent signal change in the insula and their behavioral measures of interoceptive awareness, but not between the behavioral measures only.

c) However, the associations between insular signal changes and interoceptive measures are weak and inconsistent. Why would the signal related to the heartbeat detection task show a stronger correlation with the breath-hold duration than with behavioral measures on the same task?

d) The selective report of the significant difference in the manuscript is misleading and should be rephrased. Without a conclusive indication of behavioral loadings on a latent awareness factor, I would be very cautious to interpret such correlations at all. Moreover, the absence of loadings on a shared factor should also be reflected in the Discussion which emphasizes the relevance of interoceptive awareness for the risk of suicide attempts.

---

## [Author Response]

Essential Revisions:1) Stats.a) The description of the statistical analysis in the manuscript is currently not sufficient. I was happy to look at the code, but it should not be necessary to know R to understand what exactly the authors did. Therefore, more details on the models should be provided in the Materials and methods or the supporting information.

We agree that more information describing each statistical analysis would improve the manuscript. In response to the reviewers’ questions we sought additional consultation with a statistician regarding the best way to clearly present our analyses and results. During this process, we opted to make two additional changes to our analysis approach that should increase the accuracy and methodological rigor of our study. First, we realized that by not specifying “type = marginal” in our ANOVA, R defaulted to a sequential approach, which is not the best approach for our models and can result in inconsistency between ANOVA output and LME fixed effects summary tables. We have made the appropriate correction. All ANOVA findings reported in this revision are calculated using marginal sums of squares. Additionally, we have adopted the Kenward-Roger approximation of degrees of freedom for all tests conducted using the LME. Despite having made this revision to our analysis approach, our findings are consistent with those described in our initial submission, with the minor exception of the heartbeat perception task.

We have rewritten the Materials and methods and Results sections to reflect these changes and to include more details regarding our model specifications, each analysis conducted, and the associated results. We feel that these revisions should substantially increase the clarity for readers regarding our statistical approach.

b) Relatedly, it was not obvious to me why the authors used the lme function for some analyses but not others (although it was even passed to a _gls variable) and sometimes the summary function or the anova function was used, not both.

We had initially made the decision to use a GLS for the breath-hold analyses. Although our breath-hold task consists of two trials, when we planned our analyses we decided the most parsimonious approach would be to take the mean duration of both trials and test for overall group differences, since the two trials were identical and we were not necessarily interested in performance differences from trial to trial. However, after consideration of the reviewers’ comments, we agreed that switching between analysis approaches could cause unnecessary confusion. Additionally, while a between-groups comparison may be a simpler test overall, the examination of each trial is likely more appropriate given the task design. Therefore, we have re-analyzed data from the breath-hold task using a linear mixed effects model. Our updated findings are consistent with the initial results. The linear mixed effects analyses for breath hold duration, O_2_ concentration, and CO_2_ concentration are reported in Results subsection “Breath-Hold Challenge”, and displayed in Supplementary file 1.

The reviewers note that sometimes the summary function was used without the anova function. The ANOVA function in R provides F tests for each coefficient, while “summary(model)” displays post-hoc tests describing contrasts between all levels of each factor. When only two groups are being compared (i.e., in the absence of potential interactions), the output of the fixed effects model summary and the output of the ANOVA should be identical. Additionally, when the only significant effect is a main effect of a factor with < 2 levels (e.g., a main effect of group, or a main effect of trial where there are 2 trials), the estimates of fixed effects will not provide additional information. Thus in our case it would be redundant to report both the ANOVA *F*-statistics and the fixed effects *t*-statistics. In our revised manuscript, we report overall *F*-statistics in the text, except for cases in which the fixed effects needed to be examined to clarify interactions or main effects involving three or more levels. To provide more detail on the results obtained with the models, we have also added the output of the LME summary tables to the revised manuscript supporting information section (Supplementary Files 1-3).

We have rewritten our Materials and methods and Results section with clearer language describing our analyses. Please note that in the Results section of the updated manuscript we have followed *eLife*’s guidelines to use an active voice and more engaging style. Additionally, we provide the following statement that describes our overall statistical approach (in addition to more specific details provided throughout the Materials and methods and Results section for each model):

“A marginal analysis of variance (ANOVA) was used on each LME model to examine F-tests for interactions and main effects. In the event of significant interactions, the summaries of LME fixed effects were examined to clarify which factors were driving the effect. The Kenward-Roger approximation of degrees of freedom was used for all LME analyses.”

We have submitted the revised code and statistical output in the form of an R Markdown HTML file.

c) In the manuscript, it was not clear to me which results were reported. Thus, it would be preferable to be a bit more precise in terms of the statistical descriptions and to include the output of the estimated models in the supporting information.

We agreed with the reviewers that the manuscript could be improved by providing more precision in our Materials and methods and Results sections. In making these changes in the revised manuscript, we have rewritten both sections. We have now added a paragraph in the Materials and methods section providing a description of our overall statistical approach (under the subheading “Analysis of Demographic, Behavioral, and Physiological Data”).Additionally, we have revised the language throughout the Materials and methods and Results section for each task to provide more specific details about the statistical modeling approach. To further enhance clarity, we have also added Tables displaying the exact model specification (e.g., Duration ~ Group x Trial + 1|id) and model outputs (i.e., fixed effects estimates with p-values, random effects estimates, and model fit indices). Please see Supplementary files 1-3 in our revised submission for these details.

2) Cold pressor challenge.a) The authors write: "Examination of fixed effects revealed that, while there were no significant group differences in the amount of time elapsed prior to reaching mild, moderate, and peak pain, suicide attempters kept their hands submerged in the cold water for significantly longer (18 seconds on average, SD = 25) than non-attempters after reaching peak pain (t(276)=2.8, p = 0.006, Cohen's d = .34)."b) First, does the average refer to the group difference in duration and the SD to the pooled SD across groups?

The average refers to the difference between the mean of attempters and non-attempters, and the SD is the pooled SD across groups. After giving this more consideration, we have decided that it does not make sense to report the SD here. The decision to report the group difference in duration in number of seconds is to provide a metric that is straightforward and accessible to all readers, and reporting the pooled SD here adds unnecessary confusion. Additionally, we report clearer measures of variability elsewhere (i.e., as illustrated by the error bars in Figure 2).

c) Second, if it only refers to this category, why are the degrees of freedom so high that it appears as if a fixed effect across all repeated measurements was taken for the comparison?

After receiving these reviews, we consulted with a statistician regarding our overall statistical approach, which revealed there is a debate surrounding the best approach for calculating degrees of freedom for the fixed effects in linear mixed effects models. In the revised submission, we use the Kenward-Roger approximation. This approach provides a more conservative estimate of statistical significance of fixed effects. We have clarified this in the manuscript in the paragraph that follows the subheading “Analysis of Demographic, Behavioral, and Physiological Data”.

d) Third, how was Cohen's d calculated because the study would not be large enough to find significant group differences of this magnitude if the degrees of freedom are calculated correctly (1-β = .36 for d = .34, n1 = 34, n2 = 68). This indicates that a formula was used to estimate d from t, which is only valid for non-nested t-contrasts ("simple" two-sample t-tests).

Cohen’s d was calculated using the “lme.dscore” function from the R package “EMAtools.” This function does use a formula to estimate Cohen’s d from the t-statistic [i.e., (2*t)/sqrt(df)]. In this revised submission, in lieu of Cohen’s d, we have decided to take a simpler approach and report the standardized β coefficients (i.e., slope differences in units of standard deviation of our dependent variable) in our LME summary tables. Additionally, we report Marginal R2 and Conditional R2 for the overall model. However, if the reviewers or Editors prefer that we use an alternative metric for reporting effect size, we would be happy to do so.

3) Relations amongst measures.a) At the moment, DeVille et al. are not fully capitalizing on the strength of the design provided by the complementary measures of interoceptive awareness because the correlations between the interoceptive measures are not reported.b) The authors report associations between percent signal change in the insula and their behavioral measures of interoceptive awareness, but not between the behavioral measures only.c) However, the associations between insular signal changes and interoceptive measures are weak and inconsistent. Why would the signal related to the heartbeat detection task show a stronger correlation with the breath-hold duration than with behavioral measures on the same task?d) The selective report of the significant difference in the manuscript is misleading and should be rephrased. Without a conclusive indication of behavioral loadings on a latent awareness factor, I would be very cautious to interpret such correlations at all. Moreover, the absence of loadings on a shared factor should also be reflected in the Discussion which emphasizes the relevance of interoceptive awareness for the risk of suicide attempts.

The reviewers’ concerns related to the selective reporting of the correlations in our initial submission of the manuscript was well-received. We did not intend to mislead our audience, and we agree that the practice of reporting noteworthy correlations in the main manuscript while relegating other correlations to the supplemental materials can be problematic. Therefore, we removed all text that selectively described some correlations but not others from the manuscript. Although by removing discussion of the correlations throughout the manuscript addresses the concerns described by the reviewers in point (d), it does not address the initial concern described in (a)-(c). Thus, we have added a Figure to Appendix 1 that displays the correlations across all measures from the behavioral and neuroimaging tasks, and include a brief comment at the end of the Results section indicating that “Exploratory correlations across behavioral and neuroimaging variables are reported in Appendix 1—figure 1.” Although this approach may not fully capitalize on the strength of the design, given the length of the manuscript, the breadth of the findings we are already reporting, and our lack of specific hypotheses regarding the strength and direction of the correlations across tasks, we believe that any additional focus on the correlations would detract from rather than enhance our manuscript. Additionally, given that our data is a subsample of a larger study, it may be better to analyze the relationships across interoception measures using the larger dataset that includes healthy subjects. We have added the following comment to the “Limitations and Future Directions” section:

“Although not the primary intent of our investigation, we observed several interrelationships within and across levels of analysis raising the possibility of a latent interoceptive awareness trait factor. However, these relationships were inconsistent and were not pre-specified in our hypotheses. Identification of such a latent factor would likely require additional investigation using larger samples and inclusion of individuals not meeting criteria for psychiatric disorders.”

Regarding the comment on the weak relationship between insular signal change and interoceptive measures described in (c), the reviewers raise an excellent question about the unexpected differences between the within-task and across-task correlations. It is possible that we were better able to capture variability (and therefore stronger correlations) in the measure of intensity obtained during the heartbeat detection task due to the fact a) the heartbeat perception task involved modulation of the heartbeat, and b) required subjects to provide ratings on a 0-100 scale instead of 0-6 scale (that is, more variability in the dependent variable). The relationships between these two tasks may also be worth exploring in an analysis that includes the full sample of 500 individuals, rather than a subsample of suicide attempters and matched non-attempters.

[Editors' note: we also include below the reviews that the authors received from another journal prior to submission to eLife, along with the authors’ responses.]

Reviewer #1:

Comments to authors:The manuscript “Diminished responses to bodily threat and blunted cardiac interoception in suicide Attempters" is an excellent, novel, and well-written contribution that reports physiological and self-report data that is in line with theories suggesting that suicide may root in blunted interoceptive processing or in other words increased tolerance to (impending) physiological damage. This is currently the most comprehensive study on this topic. Being a sub-study of a larger study has allowed recruiting a particularly well described sample and has offered the opportunity for individual matching of patients between suicide-attempters and non-attempters. The findings have important implications for research and clinical practice.The manuscript is well structured; language and grammar are clear and precise. The Introduction is sound, methods, sample selection as well as sample size appear appropriate and the Discussion is well balanced.

We thank the reviewer for this kind assessment of our study and manuscript.

Major issues:The only major point I did not really understand was, why the authors have chosen to match 34 suicide-attempters with 68 non-attempters, which is a sample exactly twice the size and creates un-even cell-sizes (which is associated with statistical problems).

We chose to increase the number of individuals in our reference group by using a 1:2 matching ratio in order to increase our confidence in the between-groups comparisons due to the relatively low number of suicide attempters available to us. Although unequal cell sizes can be problematic for certain statistical analyses, the analysis methods we selected are tolerant to the issues associated with unequal cell sizes (e.g., heterogeneous variances between groups).

In the sample description, I missed an overview of diagnoses and comorbidities. In addition, a bit more information on type of and reason for medication would be helpful (maybe in the appendix).

Information about medication usage and diagnostic comorbidities has been added to the Supplement. See Supplemental Table S1 for an overview of the diagnostics between groups and Supplemental Table S2 for an overview of the types of medications participants were taking.

In the main manuscript, a little more information regarding missing data would make it easier for the reader to appreciate what is reported in detail in the appendix. I would suggest shifting the sentence "non-attempter subjects (n=239) were only included for matching to suicide attempters if their data had been manually checked and they had complete observations on the behavioral and psychophysiological variables examined" to the Materials and methods section.

The relevant information has been moved to the main text of the manuscript.

I also missed a-priori power considerations.

We did not conduct or report an a-priori power analyses in the current study, due to the fact that we conducted all analyses retrospectively using a dataset that had previously been collected as part of a longitudinal naturalistic study. Additionally, given the lack of literature on behavioral and neuroimaging assessments of interoception and suicide, expected effect sizes would have been difficult to estimate beforehand. A comment on this has been added to the manuscript (subsection “Participants”) to clarify that our findings are from a retrospective analysis of a pre-existing dataset.

In the Discussion, maybe a brief remark regarding the problem of relatively low reliability of physiological measures might be useful. In particular, it should be made clear that none of the tasks applied in the study could serve for (individual) diagnosis or prognosis.

This is an important point and we have added this comment to the Discussion (“it should be noted that none of the interoceptive tasks applied in the current study have demonstrated sufficient reliability to be considered appropriate for implementation in prognostic assessments of suicidality in clinical settings*.”*).

Minor issues:Please add version/build no. of R package 'nlme'.In the sentence "Suicide attempters held their breath longer (10 seconds on average)" I'd suggest adding SD.In the figure captions, I would suggest to explain the meaning of the asterisk and report the associated significance level.

We thank the reviewer for pointing these issues out. These changes have been made in the text and in the figure captions.

Reviewer #2:

Comments to authors: The paper by Khalsa and colleagues reports on the findings of an interesting study examining the topic of diminished interoception to orthostatic threats in suicide attempters. With rising numbers of suicide deaths, identifying indices specific to attempters versus non-attempters is highly relevant and timely. The paper is well written and presents strong and interesting findings.

We appreciate the reviewer for their kind assessment of our manuscript.

I only have a few suggestions:Did the authors assess for current suicidal ideation in their sample? It seems important not only to ensure that there were no group differences but whether there is a difference between non-ideators versus ideators versus past attempters. My understanding is that research on indices between ideators versus attempters is greatly needed to better predict and target the group at greatest risk. What about current or past NSSI, particularly given that NSSI is thought to decrease pain perceptions.

Suicidal ideation was assessed in two ways, using 1) the Patient Health Questionnaire (PHQ-9) during screening, and 2) using the Columbia Suicide Severity Rating Scale (CSSR) upon entry to the Tulsa 1000 longitudinal study. Participants in the study were excluded from the study if they reported active suicidal ideation on the PHQ-9. During the study, administration of the CSSR focused on assessing lifetime suicidal ideation, not current suicidal ideation at that time. As a result, we were unable to examine current and/or recent suicidal ideation in our sample, and we recognize that this is a limitation in the Discussion. We agree that research on indices that distinguish ideators from attempters is greatly needed. To assess this limitation to the best of our ability, we have re-run our analyses after a) covarying for lifetime ideation intensity and b) dividing our non-attempter group on the basis of presence or absence of lifetime ideation. These analyses are reported in the supplement. We acknowledge that this is not a perfect way of addressing this and that further research will be needed to effectively parse out the contribution of ideation to the differences in interoception between ideators and non- ideators.

Regarding the relationship between non-suicidal self-injury (NSSI) and pain perception:

In our sample, 47% (n=16) of suicide attempters reported a history of NSSI, and 50% (n=17) of suicide attempters denied a history of NSSI. We re-analyzed the four primary interoceptive domains examined in the current study with the inclusion of an interaction term between attempt history and NSSI. We found no significant interaction between attempt history and NSSI affecting overall heartbeat perception accuracy (t(90) = 1.6, *p* = .12), breath hold duration (t(90) = 1.15, *p* = .26), cold pressor duration (*t*(85) = .16, *p* = .87), or signal in the right posterior insula during cardiac attention (*t*(90) = 1.9, *p* = .06). We have added this analysis of the role of non- suicidal self-injury and some interpretive comments to the supplement.

It would be helpful to provide a bit more support for the chosen tasks. For example, how is the sensation elicited by breath holding (suffocation) or pain using the CPT thought to relate to suicide behaviors? Why would people who can tolerate shortness of breath better be more willing to take their life? It seems a plausible pathway in NSSI were repeated cutting leads to habituation and this in terms of less pain avoidance, but the importance of the sensory perception is less clear in attempters.

Our decision to examine interoception and suicide attempt history was motivated primarily by existing theories of suicide (Klonsky and May, 2015; Van Orden et al., 2010) and the small body of experimental work on suicide attempt history and interoception (Dodd et al., 2018; Forrest, Smith, White, and Joiner, 2015; Smith, Forrest, and Velkoff, 2018). We selected available tests of interoception from our study database that had a clear relationship to vital organ systems (i.e., cardiac, respiratory) or threats to bodily integrity (i.e., pain). The ability to tolerate aversive respiratory sensations is particularly relevant, as methods involving asphyxiation account for a large and increasing proportion of death by suicide (Yau and Paschall, 2018). Although the cold pressor task may not readily resemble a specific method of suicidal behavior, it is a strong assessor of pain tolerance and has been shown to correlate well with assessments of interoceptive sensitivity (Pollatos, Füstös, and Critchley, 2012) and other modalities of pain processing (Harris and Rollman, 1983).

Regarding the comment that “It seems a plausible pathway in NSSI were repeated cutting leads to habituation and this in terms of less pain avoidance, but the importance of the sensory perception is less clear in attempters,” theoretical work on suicide has previously argued that there is are two plausible pathways between pain perception and suicide attempts, one in which pain tolerance increases over time due to repeated exposure to pain (e.g., NSSI), and one in which the individual, regardless of pain exposure, has a heightened pain threshold and/or pain tolerance (i.e., “dispositional” contributors to suicidal capability as described in the model proposed by Klonsky and May (Klonsky and May, 2015)). While it is true that repeated cutting may lead to pain habituation which, in turn, might increase one’s suicidal capacity (as described in Joiner’s theory of suicide (Van Orden et al., 2010)), a trait-level dispositional degree of pain insensitivity could also predispose individuals to engage in life-threatening and/or physically harmful behaviors (Klonsky and May, 2015). In the current study we do not attempt to dissociate whether the interoceptive differences observed in our suicide attempters is due to a dispositional or acquired factors. Our primary aim was to establish whether evidence for such interoceptive differences exists, which would lay a foundation for future studies to distinguishing the role of dispositional or acquired factors.

Since one cannot rule out that the attempt itself (e.g., via hanging) caused the diminished interoception, additionally comparing non-attempters/non-ideators to non- attempters/ideators could further elucidate interoceptive differences in suicidal versus non-suicidal individuals.

As the reviewer points out, there are two possible interpretations to our findings. The first interpretation is that individuals who possess heightened tolerance for aversive interoceptive sensations are predisposed to engage in suicidal behavior in the presence of suicidal ideation, whereas individuals who lack tolerance for aversive interoceptive sensations will be less likely to engage in suicidal behavior, even in the presence of severe ideation. Another interpretation is that there are no interoceptive differences distinguishing those with the capacity to engage in suicidal behaviors from those who lack this capacity, but that suicide attempts themselves result in diminished interoception. While we believe that the former interpretation makes the most sense given the theoretical literature on suicidal capacity (Klonsky and May, 2015; Van Orden et al., 2010), we agree with the reviewer that we cannot rule out the possibility that the act of having previously engaged in suicidal behavior caused the diminished interoception in our sample. Additionally, it will be interesting to examine whether diminished interoception at baseline predicts the later emergence of suicidal behavior from a longitudinal perspective. Since the findings we are reporting are from an ongoing naturalistic longitudinal study we plan to explore this question in the future.

The reviewer also raises the issue of comparing of attempters, ideating non-attempters, and non-ideators. We agree that the ability to distinguish between these three groups is important. We have conducted additional analyses within our current sample in an attempt to provide clarification on this issue. In these additional analyses, we re-examined our four primary interoception variables after covarying for ideation intensity, and we examined group differences after splitting our sample into three groups: suicide attempters, ideators without a history of attempts, and non-ideators (See supplement). We have also added to the Discussion of this limitation in the main body of the manuscript.

Why was the breath-holding test repeated?

The breath holding test was repeated and averaged in order to increase our confidence in the measure through multiple observations, matching the approach of previous studies utilizing this test (Asmundson and Stein, 1994).

"greater expired O2 following breath hold" – should this say "lesser"?

This typo was corrected.

It is not clear why the authors report the findings on CO2 and O2. Of course, one would expect that breath holding time goes along with higher CO2 and (a small but non-clinical) decrease in O2 – in my mind that is not a result but just a manipulation check.

This is more than a manipulation check. Measurement of CO2 levels provided physiological evidence of suicide attempters’ greater ability to tolerate acute hypercapnic body states. Of course, based on gas exchange laws the O2 and CO2 levels were expected to be reciprocally related, and in that regard, including both could be viewed as a form of manipulation check. Leaving both results allows readers to assess both possibilities.

Reviewer #3:

Comments to authors: This interesting and potential important paper further strengthens previous studies suggesting a link between interoceptive processes and suicidality. Its strengths are twofold, firstly replicating previous findings of an association between self report interoceptive measures and recent (last 5 years) suicide plans, but more importantly substantiating these with more objective experimental measures of interception and their linked brain regions i.e. insula.

We thank the reviewer for this assessment.

I only have minor comments:

1) Many of the interception measures have either an implicit or explicit visceral pain component – it would be nice to be able to dissociate visceral pain (e.g. cold water immersion) from more somatic types of pain (e.g. Von Frey hairs) or alternately somatic sensory sensitivity to try and clarify whether suicidality is linked to impaired pain or somatic perception per se (nociception) or more specifically to interoceptive processes. It would be useful to at least raise this issue in the Discussion. e.g. [1][1] Pollatos O, Füstös J, and Critchley HD (2012). On the generalised embodiment of pain: how interoceptive sensitivity modulates cutaneous pain perception. Pain, 153 (8), 1680-6

We agree with the reviewer and have added a comment to the Discussion and is replicated below:

“The current study employed an imprecise, albeit commonly employed, measure of pain perception. It would be advantageous to be able to clarify whether suicidal action is differentially linked to impaired processing of visceral pain signals, which tend to be poorly localized, often referred to somatic structures, and produce strong autonomic and affective responses, as opposed to somatic pain signals, which tend to be discretely localized to somatic structures and produce more variable autonomic and affective responses (Sikandar and Dickenson, 2012). Beyond stimulating visceral and somatic pain processing via different neuroanatomical pathways, it would be helpful to evaluate the degree to which altered pain responding was directly driven by differences in nociception per se as opposed to indirectly modulated by differences in interoceptive processing (Pollatos et al., 2012).”

2) Previous authors have shown/ not shown an association between interception and personality features particularly emotionally unstable PD [2]. Do personality features mediate moderate link between interoception and suicidality in this cohort where personality measures were recorded?[2] Hart N, McGowan J, Minati L, Critchley HD. Emotional regulation and bodily sensation: interoceptive awareness is intact in borderline personality disorder. J Personal Disord. 2013;27(4):506-18.[3] Müller LE, Schulz A, Andermann M, Gäbel A, Gescher DM, Spohn A, et al. Cortical representation of afferent bodily signals in borderline personality disorder: neural correlates and relationship to emotional dysregulation. JAMA psychiatry. 2015;72(11):1077-86.

We unfortunately do not have personality measures related to emotional stability/instability assessed in the current study. However, this raises an interesting question, and we have discussed the possibility further in the Supplement.

Reviewer #4:

This study examines several indices of interoceptive awareness in psychiatric patients with histories of suicide attempts (n=34) and a psychiatric comparison group with similar levels of psychopathology (n=68). The topic of the study is timely, as the field knows little about what factors facilitate transition from suicidal thoughts to acts, and interoceptive awareness may be one of them. However, problems with sample size and design I think greatly limit what information can be gleaned from this study's results.Sample size.It seems sample size is insufficient for accurate estimation of population parameters, which not only means that real effects can be missed, but that seemingly 'significant' effects can be false positives. I worry this is a substantial rather than small problem when it comes to interpreting results: there are so many different statistical comparisons reported between the two groups, but by my calculation power for even a *single* statistical test is at most around 70%.

The reviewer raises an important point regarding the concern of false positives when conducting analyses within smaller samples. In the current study, we assessed our subjects across multiple distinct, yet related domains (i.e., heartbeat tapping, insula activation during cardiac interoceptive attention, breath holding, cold pressor performance, and self-report indices of interoception). While we did not produce statistically significant results for every analysis conducted (nonsignificant results are reported in the main text of the manuscript and the supplement), we observed a significant difference between groups on cold pressor and breath hold task performance as well as behavioral and neuroimaging assessments of cardiac interoception. We argue that erroneously receiving false positives across multiple measures of interoception is improbable, particularly given that 1) differences between groups were consistent across tasks and in the same direction (i.e., suicide attempters with blunted interoception relative to non-attempters), 2) we observed similar effect sizes across tasks that were in the moderate range (i.e., reducing suspicion for inflated effect sizes associated with small samples), and 3) our findings are supported by pre-existing theoretical work.

It should be noted that our sample size (N=102) is larger than what has been typically used in neuroimaging studies. The median sample size in a published neuroimaging study has been reported as 28 (Poldrack et al., 2017) and 33 subjects (Yeung, 2018), with nearly 95% of studies reporting a sample size of <100 subjects (Yeung, 2018). Additionally, with our data coming from a larger longitudinal study, we will have the opportunity to replicate our findings as more data are collected.

Design.The potential role of suicide capacity highlighted in the manuscript is to facilitate “transitioning from suicidal ideation to action”. Thus, the group with attempts should have been matched to a group with ideation but no attempts. In contrast, matching on psychopathology is an inappropriate control for this research question, and at best is a rough and unsatisfactory proxy for ideation. Table 1 does report differences between the two groups on a single self-report item assessing suicidal ideation showing a moderate elevation in ideation among the attempter group (d is approximately.3). But the manuscript indicates the Columbia Suicide Severity Rating Scale was administered, and this provides an assessment of suicidal ideation that could be the basis for matching – why would this suicide ideation assessment be ignored when the difference between ideation and attempts is the focus of the manuscript? Matching the groups on ideation is critically important because all attempters have ideation; thus, in a design comparing attempters to non-attempters, a variable that correlates with ideation can masquerade as a correlate of attempts. We only know if a variable correlates with attempts above and beyond its relationship to ideation if ideation is explicitly and carefully controlled for. This design feature is necessary to address the manuscript’s focus on suicide capacity, since the whole point of capacity is that it may contribute to the difference between ideation alone and ideation that progresses to attempts.

We agree that matching on psychopathology alone would be an inappropriate and unsatisfactory proxy for ideation. However, we did not intend to use psychiatric symptom severity as a proxy for ideation. Rather, our goal was to match for illness markers that could independently impact our interoception variables. Anxiety, depression, eating disorders, and substance use disorders have all been previously associated with differences in interoception [e.g., response to cardiac and respiratory perturbation in anxiety disorders (Pohl et al., 1988; Van den Hout et al., 1987); differences in the insula BOLD signal during interoceptive attention in depression (Avery et al., 2014) and eating disorders (Kerr et al., 2016); cold pressor performance differences in individuals with depression (Schwier, Kliem, Boettger, and Bär, 2010) and substance use disorders (Pud, Cohen, Lawental, and Eisenberg, 2006); aberrant insula activation during an inspiratory breathing load task in individuals with a substance use disorder (Stewart et al., 2014); heightened sensitivity to heartbeat sensations (Khalsa et al., 2015) and aberrant insula response to respiratory perturbation in anorexia nervosa (Berner et al., 2018)]. By matching on psychopathology, we increased the likelihood that any differences observed on our measures of interoception were related to suicide attempt history rather than differences in psychopathology between groups. To clarify this in the manuscript, we have added text to the Discussion section (“We demonstrated these differences in a well-characterized sample of participants with similar levels of psychiatric symptoms, increasing the likelihood that the participants’ history of suicidal behavior was the driving force behind the differences observed in interoception, rather than varying features of psychopathology between groups”).

To address concerns about distinguishing ideators and attempters, we report on additional analyses in the Supplement showing that the relationship between interoception and suicide attempts is maintained even after covarying for suicidal ideation. We have also expounded on the limitations that we face by broadly comparing suicide attempters and non- attempters in the Discussion section of our manuscript, and call for additional research that will better enable us to examine the role of interoception in attempters, ideators, and non-ideators. We do believe, however, that the present examination of the role of interoception in suicide represents the strongest starting point for this emerging body of research to date.

I think there’s a mistake in Table 1.% Female is listed as 70% but the n is listed as just 4, which is well under 10%

We thank the reviewer for pointing out this error. The number of females in this cell should have been 47, not 4. Table 1 has been revised accordingly.

References:

Asmundson, G. J., and Stein, M. B. (1994). Triggering the false suffocation alarm in panic disorder patients by using a voluntary breath-holding procedure. *The American journal of psychiatry*.

Avery, J. A., Drevets, W. C., Moseman, S. E., Bodurka, J., Barcalow, J. C., and Simmons, W. K. (2014). Major depressive disorder is associated with abnormal interoceptive activity and functional connectivity in the insula. *Biol Psychiatry, 76*(3), 258-266. doi:10.1016/j.biopsych.2013.11.027

Berner, L. A., Simmons, A. N., Wierenga, C. E., Bischoff-Grethe, A., Paulus, M. P., Bailer, U.,. Kaye, W. H. (2018). Altered interoceptive activation before, during, and after aversive breathing load in women remitted from anorexia nervosa. *Psychological medicine, 48*(1), 142-154.

Dodd, D. R., Smith, A. R., Forrest, L. N., Witte, T. K., Bodell, L., Bartlett, M.,... Goodwin, N. (2018). Interoceptive deficits, nonsuicidal self-injury, and suicide attempts among women with eating disorders. *Suicide Life Threat Behav, 48*(4), 438-444.

Forrest, L. N., Smith, A. R., White, R. D., and Joiner, T. E. (2015). (Dis)connected: An examination of interoception in individuals with suicidality. *J Abnorm Psychol, 124*(3), 754-763.

Harris, G., and Rollman, G. B. (1983). The validity of experimental pain measures. *Pain, 17*(4), 369-376.

Kerr, K. L., Moseman, S. E., Avery, J. A., Bodurka, J., Zucker, N. L., and Simmons, W. K. (2016). Altered Insula Activity during Visceral Interoception in Weight-Restored Patients with Anorexia Nervosa. *Neuropsychopharmacology, 41*(2), 521-528. doi:10.1038/npp.2015.174

Khalsa, S. S., Craske, M. G., Li, W., Vangala, S., Strober, M., and Feusner, J. D. (2015). Altered interoceptive awareness in anorexia nervosa: effects of meal anticipation, consumption and bodily arousal. *International Journal of Eating Disorders, 48*(7), 889-897.

Klonsky, E. D., and May, A. M. (2015). The three-step theory (3ST): A new theory of suicide rooted in the “ideation-to-action” framework. *International Journal of Cognitive Therapy, 8*, 114-129. doi:https://doi.org/10.1521/ijct.2015.8.2.114

Pohl, R., Yeragani, V. K., Balon, R., Rainey, J. M., Lycaki, H., Ortiz, A.,... Weinberg, P. (1988). Isoproterenol-induced panic attacks. *Biological psychiatry, 24*(8), 891-902.

Poldrack, R. A., Baker, C. I., Durnez, J., Gorgolewski, K. J., Matthews, P. M., Munafò, M. R.,... Yarkoni, T. (2017). Scanning the horizon: towards transparent and reproducible neuroimaging research. *Nature Reviews Neuroscience, 18*(2), 115.

Pollatos, O., Füstös, J., and Critchley, H. D. (2012). On the generalised embodiment of pain: how interoceptive sensitivity modulates cutaneous pain perception. *Pain, 153*(8), 1680-1686.

Pud, D., Cohen, D., Lawental, E., and Eisenberg, E. (2006). Opioids and abnormal pain perception: New evidence from a study of chronic opioid addicts and healthy subjects. *Drug and alcohol dependence, 82*(3), 218-223.

Schwier, C., Kliem, A., Boettger, M. K., and Bär, K.-J. (2010). Increased cold-pain thresholds in major depression. *The Journal of Pain, 11*(3), 287-290.

Sikandar, S., and Dickenson, A. H. (2012). Visceral pain–the ins and outs, the ups and downs.

*Current opinion in supportive and palliative care, 6*(1), 17.

Smith, A. R., Forrest, L. N., and Velkoff, E. (2018). Out of touch: Interoceptive deficits are elevated in suicide attempters with eating disorders. *Eat Disord, 26*(1), 52-65.

Stewart, J. L., May, A. C., Poppa, T., Davenport, P. W., Tapert, S. F., and Paulus, M. P. (2014). You are the danger: attenuated insula response in methamphetamine users during aversive interoceptive decision-making. *Drug and alcohol dependence, 142*, 110-119.

Van den Hout, M. A., der Molen, V., Margo, G., Griez, E., Lousberg, H., and Nansen, A. (1987). Reduction of CO₂-induced anxiety in patients with panic attacks after repeated CO₂ exposure. *The American journal of psychiatry*.

Van Orden, K. A., Witte, T. K., Cukrowicz, K. C., Braithwait, S., Selby, E. A., and Joiner, T. E. (2010). The interpersonal theory of suicide. *Psychol Rev, 117*(2), 575-600. doi:doi:10.1037/a0018697

Yau, R. K., and Paschall, M. J. (2018). Epidemiology of asphyxiation suicides in the United States, 2005–2014. *Injury epidemiology, 5*(1), 1.

Yeung, A. W. (2018). An updated survey on statistical thresholding and sample size of fMRI studies. *Frontiers in human neuroscience, 12*, 16.